# Learning Relevant Contextual Variables Within Bayesian Optimization

**Julien Martinelli**[*1]    **Ayush Bharti**[2]    **Armi Tiihonen**[3]    **S. T. John**[2]    **Louis Filstroff**[4]    **Sabina J. Sloman**[5]

**Patrick Rinke**[3]                    **Samuel Kaski**[2,5]

[1]Inserm Bordeaux Population Health, Vaccine Research Institute, Université de Bordeaux, Inria Bordeaux Sud-ouest, France
[2]Department of Computer Science, Aalto University, Helsinki, Finland
[3]Department of Applied Physics, Aalto University, Helsinki, Finland
[4]Univ. Lille, CNRS, Centrale Lille, UMR 9189 CRIStAL, F-59000 Lille, France
[5]Department of Computer Science, University of Manchester, Manchester, United Kingdom

## Abstract

Contextual Bayesian Optimization (CBO) efficiently optimizes black-box functions with respect to design variables, while simultaneously integrating *contextual* information regarding the environment, such as experimental conditions. However, the relevance of contextual variables is not necessarily known beforehand. Moreover, contextual variables can sometimes be optimized themselves at an additional cost, a setting overlooked by current CBO algorithms. Cost-sensitive CBO would simply include optimizable contextual variables as part of the design variables based on their cost. Instead, we adaptively select a subset of contextual variables to include in the optimization, based on the trade-off between their *relevance* and the additional cost incurred by optimizing them compared to leaving them to be determined by the environment. We learn the relevance of contextual variables by sensitivity analysis of the posterior surrogate model while minimizing the cost of optimization by leveraging recent developments on early stopping for BO. We empirically evaluate our proposed Sensitivity-Analysis-Driven Contextual BO (`SADCBO`) method against alternatives on both synthetic and real-world experiments, together with extensive ablation studies, and demonstrate a consistent improvement across examples.

## 1   INTRODUCTION

Bayesian optimization (BO) is a sample-efficient black-box optimization method, typically used when the objective function is too expensive to optimize directly [Garnett, 2023]. Given an objective function that can be evaluated pointwise over a set of *design variables*, BO combines surrogate mod-

eling with a pre-specified policy of evaluation over the design space (the so-called acquisition function) to efficiently locate the global optimum of the function. BO has been especially useful in automatic discovery of materials [Zhang et al., 2020], molecules [Fang et al., 2021], and pharmaceutical compounds [Gómez-Bombarelli et al., 2018, Korovina et al., 2020]—problem domains in which evaluating the performance of a candidate depends on a costly experiment.

Despite the success of BO and its recent algorithmic advancements, open challenges remain for its practical use. A key implicit assumption in vanilla BO is that the objective function only depends on the design variables. This assumption is violated in many practical scenarios, wherein various *uncontrolled* environmental factors and experimental settings, referred to as *contextual variables* [Krause and Ong, 2011, Kirschner et al., 2020, Arsenyan et al., 2023], also affect the objective function. For instance, ambient humidity was found to influence the experiments in robot-assisted material design [Nega et al., 2021], such that the best compound differed with humidity conditions. Moreover, in practice, the domain experts themselves might not know *a priori* which contextual variables are relevant, and would observe their confounding effect only during the course of the optimization process. Therefore, it is critical to identify the contextual variables that significantly affect the objective function, not only to achieve the highest optimization results, but also for the practitioners to reliably reproduce experimental results.

To deal with the uncertainty related to the contextual variables, variants of BO have been developed. In particular, Krause and Ong [2011] introduced the Contextual Bayesian optimization (CBO) framework, which uses the uncontrollable contextual information known *before* the experiment, like current environmental conditions, to enhance the surrogate model. Alternatively, several works have proposed to alter the simple optimization objective to make it robust in some sense, such as by taking the expectation with respect to the contextual variables [Toscano-Palmerin and Frazier, 2022], or considering distributionally-robust scenar-

---

*Work done while at Aalto University.

ios [Bogunovic et al., 2018, Kirschner et al., 2020]. However, these works consider a different setup than the original CBO framework, as contextual information is only revealed *after* the design has been sent for experiment, not before. Besides this distinction, in some applications, contextual variables *can* be controlled, and therefore set to values they may be unlikely to take during passing observation. Such variables are, for instance, synthesis conditions of material samples, including sintering temperature or the used solvents. Certain environmental conditions like room temperature or ambient humidity are also "principally" controllable during the course of an experiment [Higgins et al., 2021, Nega et al., 2021]. Nevertheless, whether their inclusion as optimization variables is relevant or not may not be straightforward to predict [Abolhasani and Brown, 2023]. Moreover, optimizing over all the potentially relevant contextual variables can improve BO performance, but this process can be costly, thus invoking a cost-versus-efficiency trade-off.

**Contributions.** In this paper, we extend the CBO framework to settings in which the relevance of contextual variables is (i) not known beforehand, and (ii) can be optimized, but at some cost. We propose a Sensitivity-Analysis-Driven CBO (SADCBO) algorithm for the simultaneous identification and optimization of relevant contextual variables. SADCBO leverages recent advances in sensitivity-analysis-driven variable selection [Sebenius et al., 2022] and early stopping criteria for BO [Ishibashi et al., 2023]. We emphasize that SADCBO combines the *contextual observational* setting, where the context information is only observed, and the *contextual optimization* setting, where contextual variables can be optimized (similar to design variables), into a sequential algorithm. In effect, SADCBO provides a way to navigate the following tradeoff: should contextual variables be taken *as is* at no cost, or should they be steered outside of their observational distribution in order to provide more information about the objective, at a cost? We evaluate the performance of SADCBO, comparing against methods from the CBO and high-dimensional BO literature, on both synthetic and real-world cases.

## 2 CONTEXTUAL BAYESIAN OPTIMIZATION (CBO)

The CBO framework [Krause and Ong, 2011] deals with a black-box function $f : \mathcal{X} \times \mathcal{Z} \to \mathbb{R}$ defined on the joint space of both the *design variables* $\mathcal{X} \subset \mathbb{R}^d$ and *contextual variables* $\mathcal{Z} \subset \mathbb{R}^c$. We assume that we get noisy evaluations of $f$, that is, we observe the output $y = f(\mathbf{x}, \mathbf{z}) + \varepsilon$ with $\varepsilon \sim \mathcal{N}(0, \sigma_{\text{noise}}^2)$. A Gaussian process (GP) prior [Rasmussen and Williams, 2006] is placed on $f$; with the notation $\mathbf{v} = [\mathbf{x}, \mathbf{z}]$, we write $f(\mathbf{v}) \sim \mathcal{GP}(0, k(\mathbf{v}, \mathbf{v}'))$.

A GP is a stochastic process fully characterized by its mean function (taken here to be zero) and its kernel $k(\mathbf{v}, \mathbf{v}') = \text{cov}[f(\mathbf{v}), f(\mathbf{v}')]$. This implies that for any finite-dimensional collection of inputs $[\mathbf{v}_1, \ldots, \mathbf{v}_t]$, the function values $\mathbf{f} = [f(\mathbf{v}_1), \ldots, f(\mathbf{v}_t)]^\top \in \mathbb{R}^t$ follow a multivariate normal distribution $\mathbf{f} \sim \mathcal{N}(\mathbf{0}, \mathbf{K})$, where $\mathbf{K} = (k(\mathbf{v}_i, \mathbf{v}_j))_{1 \leq i, j \leq t}$ is the kernel matrix. Given a dataset $\mathcal{D}_t = \{(\mathbf{x}_i, \mathbf{z}_i, y_i)\}_{i=1}^t = \{(\mathbf{v}_i, y_i)\}_{i=1}^t$, the posterior distribution of $f(\mathbf{v})$ given $\mathcal{D}_t$ is Gaussian, with analytical expressions for the mean $\mu_t(\mathbf{v}|\mathcal{D}_t)$ and variance $\sigma_t^2(\mathbf{v}|\mathcal{D}_t)$.

In the CBO setting, we first observe the context variables, and then choose the design variables accordingly. More precisely, at iteration $t+1$, a context vector $\mathbf{z}_{t+1}$ is observed, assumed to have been drawn from an unknown distribution $p(\mathbf{z})$, and the optimal design $\mathbf{x}_{t+1}^\star$ is such that

$$\mathbf{x}_{t+1}^\star = \arg\max_{\mathbf{x} \in \mathcal{X}} f(\mathbf{x}, \mathbf{z}_{t+1}). \tag{1}$$

Given $\mathbf{z}_{t+1}$ and the previous $t$ observations $\mathcal{D}_t$, the next candidate design $\mathbf{x}_{t+1}$ is selected using the Upper Confidence Bound (UCB) acquisition function $\alpha$ [Srinivas et al., 2012]:

$$\mathbf{x}_{t+1} = \arg\max_{\mathbf{x} \in \mathcal{X}} \alpha(\mathbf{x}, \mathbf{z}_{t+1}|\mathcal{D}_t)$$
$$= \mu_t(\mathbf{x}, \mathbf{z}_{t+1}|\mathcal{D}_t) + \beta_t^{1/2} \sigma_t(\mathbf{x}, \mathbf{z}_{t+1}|\mathcal{D}_t), \tag{2}$$

for a sequence $(\beta_t)_{t \geq 1}$. This incurs a design cost $\lambda_\mathbf{x}$.

**Extending the CBO problem setup.** We extend the problem setting of CBO in two ways. Firstly, we assume that only a subset of the contextual variables truly affect $f$. Let $\mathbf{z} = [z^{(1)}, \ldots, z^{(c)}]$ be the vector of all contextual variables. For any set $J$ belonging to the power set of $\{1, \ldots, c\}$, denote by $\mathbf{z}^{(J)} \in \mathbb{R}^{|J|}$ the vector of reduced dimension whose variables are indexed by $J$. For instance, if $J = \{1, 3\}$, then $\mathbf{z}^{(J)} = [z^{(1)}, z^{(3)}]$. We assume that there exists a set $J^\star$, where $|J^\star| \ll c$, such that $f(\mathbf{x}, \mathbf{z}) = f(\mathbf{x}, \mathbf{z}^{(J^\star)}) \; \forall (\mathbf{x}, \mathbf{z})$. Secondly, we include the possibility of setting the value of any of the contextual variables at some cost over and above the usual design query cost $\lambda_\mathbf{x}$. This means that for all $j \in \{1, \ldots, c\}$, the context variable $z^{(j)}$ can be optimized at a cost $\lambda_j$. To be able to control each contextual variable, we must also assume their independence: $p(\mathbf{z}) = \prod_{j=1}^c p(z^{(j)})$. With these additional assumptions, we aim to maximize the function $f$ in a cost-efficient manner, while identifying the optimal set $J^\star$. This provides the user with a comprehensive summary of the relevant contextual variables found through optimization, thus ensuring reproducibility and explainability. Unlike CBO, the ability to control contextual variables allows us to judge whether or not one should optimize contextual variables to learn more about the objective (albeit at a cost), or if the current sampled context is already informative enough. Specifically, we aim to maximize the objective

$$(\mathbf{x}_{t+1}^\star, \mathbf{z}_{t+1}^\star) = \arg\max_{(\mathbf{x}, \mathbf{z}_{t+1}^{(J^\star)}) \in \mathcal{X} \times \prod_{j \in J^\star} \mathcal{Z}_j} f(\mathbf{x}, \mathbf{z}_{t+1}) \tag{3}$$

where, for all $j \in J^\star$, we optimize $z_{t+1}^{(j)}$ at cost $\lambda_j$, and all other elements $j' \in \{1, \dots, c\} \setminus J^\star$ of $\mathbf{z}_{t+1}$ remain at their values sampled from the environment ($z_{t+1}^{(j')} \sim p(z^{(j')})$).

## 3 METHODOLOGY

To solve the extended CBO problem introduced in Section 2, we identify relevant contextual variables, building on a variable selection technique from the GP literature [Sebenius et al., 2022]. Section 3.1 describes our adaptation of this method to the optimization setting, by restricting the dataset to high function values. Section 3.2 then presents our sequential algorithm SADCBO, which employs the adapted variable selection method in solving the optimization problem. A flowchart summarizing the proposed method can be found in Figure S1.

### 3.1 VARIABLE SELECTION FOR CBO VIA SENSITIVITY ANALYSIS

To handle the presence of contextual variables that can be optimized, one approach is to include them in the design space. However, such a strategy can be infeasible when their relevance is not known *a priori* and domain experts can only provide a candidate set of *potentially* relevant contextual variables. Indeed, this leads to an exponential expansion of the search space, while at the same time increasing the cost of optimization. In such cases, it is crucial to identify the *relevant* contextual variables, i.e., to find (a good approximation to) the optimal set $J^\star$. This not only allows us to optimize the function more efficiently but also provides additional insights about the experiment to the domain experts.

To approximate the optimal set $J^\star$, we include those contextual variables that are most relevant for identifying the optimum, which we estimate using sensitivity analysis. Specifically, we adapt the Feature Collapsing (FC) method [Sebenius et al., 2022]. The FC method perturbs training points (namely, by setting one feature to zero), and measures the induced shift in the posterior predictive distribution in terms of KL divergence. Given a dataset $\mathcal{D}_t = \{(\mathbf{x}_i, \mathbf{z}_i, y_i)\}_{i=1}^t$, the relevance $r_{i,j}$ on the $i^{\text{th}}$ sample of the $j^{\text{th}}$ contextual variable $z_i^{(j)}$ is computed as

$$r_{i,j} = \mathrm{KL}\left(p(y_\star|\mathbf{x}_i, \mathbf{z}_i, \mathcal{D}_t) || p(y_\star|\mathbf{x}_i, \mathbf{z}_i \odot \boldsymbol{\xi}[j], \mathcal{D}_t)\right), \quad (4)$$

where $\boldsymbol{\xi}[j] = [\xi^{(1)}, \dots, \xi^{(c)}]$ is a vector so that $\xi^{(j)} = 0$, and $\xi^{(j')} = 1$ for $j' \neq j$, and $\odot$ is the element-wise multiplication. The relevance score of the $j^{\text{th}}$ contextual variable is then computed as an average over $\mathcal{D}_t$:

$$\mathrm{FC}_{\mathcal{D}_t}(j) = \frac{1}{|\mathcal{D}_t|} \sum_{i=1}^{|\mathcal{D}_t|} \left( \frac{r_{i,j}}{\sum_{j'=1}^c r_{i,j'}} \right). \quad (5)$$

The FC scores obtained in this manner reveal the variables that are relevant for predicting the output *across* $\mathcal{D}_t$. As our

goal is to *maximize* $f$, we are interested in identifying contextual variables that are relevant for *high* function values. Hence, we adapt Equation (5) to the BO setting by modifying the dataset over which the scores are averaged. We use information about high function values from two different sets: (1) The subset $\mathcal{D}^{\gamma_t}$ associated with the highest output values observed so far:

$$\mathcal{D}_t^{\gamma_t} = \{(\mathbf{x}_i, \mathbf{z}_i, y_i) \in \mathcal{D}_t \mid y_i/y_{\text{best}} \geq \gamma_t\}, \quad (6)$$

where $y_{\text{best}} = \max_{1 \leq i \leq t} y_i$ is the current observed maximum. For example, using $\gamma_t = 0.8 \ \forall t$ would yield a $\mathcal{D}_t^{\gamma_t}$ that consists of the highest $20\%$ observations so far. (2) We select a batch of $Q$ points $\mathcal{D}_t^Q := \{(\mathbf{x}_q^\star, \mathbf{z}_{t+1})\}_{q=1}^Q$ that are promising given the next context $\mathbf{z}_{t+1}$:

$$\{\mathbf{x}_q^\star\}_{q=1}^Q = \underset{\{\mathbf{x}_q\}_{q=1}^Q \in \mathcal{X}^Q}{\arg\max} \ \alpha^{\text{Batch}}(\{(\mathbf{x}_q, \mathbf{z}_{t+1})\}_{q=1}^Q | \mathcal{D}_t), \quad (7)$$

where $\alpha^{\text{Batch}}$ denotes a batched version of the acquisition function $\alpha$ such as $Q$-UCB for UCB [Wilson et al., 2017]. We use the union $\mathcal{D}_t^{\text{BO}} = \mathcal{D}_t^{\gamma_t} \cup \mathcal{D}_t^Q$ as our dataset for FC. Therefore, we compute $\mathrm{FC}_{\mathcal{D}_t^{\text{BO}}}$ based on Equation (5). The importance of working with $\mathcal{D}_t^{\text{BO}}$ instead of $\mathcal{D}_t$ is illustrated in Figure 1 on a toy example.

We successively select the indices of the contextual variables with the highest FC scores until their cumulative FC score exceeds some chosen threshold $\eta \in [0, 1]$, meaning that the selected variables explain the fraction $\eta$ of the output sensitivity amongst all contextual variables. Let $J_\eta$ denote the set of indices of the selected contextual variables. We train a GP surrogate based on $\{(\mathbf{x}_i, \mathbf{z}_i^{(J_\eta)}, y_i)\}_{i=1}^t$ and can select a new design through maximization of the acquisition function $\alpha$:

$$\mathbf{x}_{t+1} = \underset{\mathbf{x} \in \mathcal{X}}{\arg\max} \ \alpha(\mathbf{x}, \mathbf{z}_{t+1}^{(J_\eta)} | \mathcal{D}_t). \quad (8)$$

Note that other measures of variable relevance could have been used, e.g., the method proposed by Spagnol et al. [2019] based on maximum mean discrepancy [Gretton et al., 2012]. We found FC to perform better (see Section 5.2).

### 3.2 SENSITIVITY-ANALYSIS-DRIVEN CBO (SADCBO)

Building on top of the variable selection method discussed in Section 3.1, we now present SADCBO, a sequential method for performing BO in the presence of irrelevant contextual variables. SADCBO proceeds in two phases.

In the first, *observational* phase, we choose to only observe the values of the contextual variables without optimizing over them. This ensures that we do not waste budget optimizing the contextual variables when their relevance is computed based on a limited amount of data, and hence can be noisy. We select the contextual variables based on their FC relevance and then use vanilla CBO as described

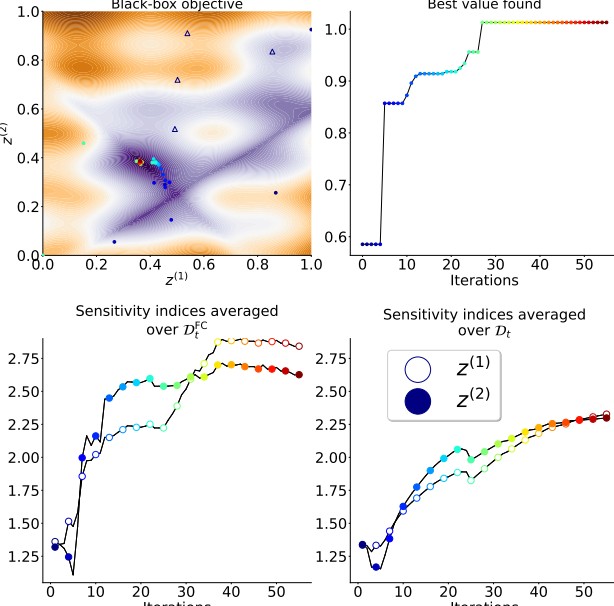

Figure 1: **Sensitivity analysis on $\mathcal{D}_t^{\text{BO}}$ characterizes variable importance at the optimum faster than $\mathcal{D}_t$.** *Top left*: 2D black-box objective together with the queries produced along a BO trajectory. Initial samples are represented by empty dark-colored triangles, newly obtained samples as dots with an increasingly lighter color. *Top right*: Best value found during the optimization trial. *Bottom left*: Sensitivity indices for $z^{(1)}$ and $z^{(2)}$ averaged over $\mathcal{D}_t^{\text{BO}}$. As we converge to the optimum, $\mathcal{D}_t^{\text{BO}}$ mainly involves samples close to the optimum, leading to a different variable relevance ranking (iteration 30 to the end; $z^{(1)}$ is more relevant) compared to the early iterations (10 to 30; $z^{(2)}$ is more relevant). *Bottom right*: Sensitivity indices computed on the whole dataset $\mathcal{D}_t$ do not converge as quickly and do not capture the shift in relevance close to the optimum.

198 in Section 2 to optimize the design variables. Thus, in this
199 phase, we leverage the available contextual information to
200 guide design selection.

201 In the early stage of the optimization, cheap queries where
202 contextual variables are not optimized still provide a con-
203 siderable amount of information. The information gained
204 from purely observing contextual variables will, however,
205 saturate at some point, leading to diminishing simple regret
206 differences. At this point, it becomes necessary to pay the
207 higher price to control more dimensions of the input space.
208 This motivates the introduction of a second phase, in which
209 contextual variables can have their values arbitrarily set,
210 through optimization.

211 In the second, contextual *optimization* phase, we optimize
212 the contextual variables selected at each iteration based on
213 their FC relevance. As optimizing a context variable $z^{(j)}$ is
214 associated with a cost $\lambda_j$, we modify the FC relevance in

Equation (5): 215

$$\tilde{\text{FC}}_{\mathcal{D}_t}(j) = \text{FC}_{\mathcal{D}_t}(j)/\lambda_j \quad (9)$$

Our variable selection criterion can then be interpreted as the 216
degree of sensitivity *per unit cost*. This allows SADCBO to 217
automatically trade off a variable's potential to greatly affect 218
the optimum with the associated optimization cost. As be- 219
fore, once the contextual variables $\mathbf{z}^{(J_\eta)}$ have been selected, 220
we train a GP surrogate using $\{(\mathbf{x}_i, \mathbf{z}_i^{(J_\eta)}, y_i)\}_{i=1}^t$ and select 221
the next design and contextual variables to query as 222

$$(\mathbf{x}_{t+1}, \mathbf{z}_{t+1}^{(J_\eta)}) = \underset{(\mathbf{x}, \mathbf{z}^{(J_\eta)}) \in \mathcal{X} \times \prod_{j \in J_\eta} \mathcal{Z}_j}{\arg\max} \alpha(\mathbf{x}, \mathbf{z}^{(J_\eta)} | \mathcal{D}_t).$$

$$(10)$$

In effect, $J_\eta$ represents our approximation for $J^\star$ as intro- 223
duced in Equation (3). Note that our acquisition function is 224
not cost-weighted, as cost-weighted acquisition functions 225
can dramatically underperform [Lee et al., 2021], specifi- 226
cally for non-continuous cost models. Including the cost at 227
the model selection level avoids this issue. 228

**Switching from observational to optimization phase.** 229
We employ the criterion proposed by Ishibashi et al. [2023] 230
for determining the stopping time in BO. Using this cri- 231
terion, we detect the point at which the optimization gain 232
based on purely observing the contextual variables dimin- 233
ishes, following which the contextual optimization phase 234
begins. We now briefly describe the details of this switching 235
criterion. 236

With $\mathbf{v} = [\mathbf{x}, \mathbf{z}]$, let $\mathbf{v}_t^\star = \arg\max_{\mathbf{v} \in \mathcal{D}_t} f(\mathbf{v})$ be the
current best candidate point in the dataset up to time
$t$. Denoting $f^\star := \max_{\mathbf{v} \in \mathcal{V}} f(\mathbf{v})$, let $R_t = f^\star - \mathbb{E}_{\hat{f} \sim p(f|\mathcal{D}_t)}[\max_{\mathbf{v} \in \mathcal{V}} \hat{f}(\mathbf{v})]$ be the expected minimum sim-
ple regret. Then, with probability $1 - \delta$, $\Delta R_t = |R_t - R_{t-1}|$
can be upper bounded by $\Delta\tilde{R}_t$ with

$$\Delta\tilde{R}_t = v(\phi(g) + g\Phi(g)) + |\Delta\mu_t^\star|$$
$$+ \kappa_{\delta,t-1}\sqrt{\frac{1}{2}\text{KL}(p(f|\mathcal{D}_t)||p(f|\mathcal{D}_{t-1}))}, \quad (11)$$

where $\phi(\cdot)$ and $\Phi(\cdot)$ are the p.d.f. and c.d.f. of a stan- 237
dard Gaussian distribution, respectively, $\Delta\mu_t^\star := \mu_t(\mathbf{v}_t^\star) -$ 238
$\mu_{t-1}(\mathbf{v}_{t-1}^\star), v := \sqrt{\sigma_t^2(\mathbf{v}_t^\star) - 2\Sigma_t(\mathbf{v}_t^\star, \mathbf{v}_{t-1}^\star) + \sigma_t^2(\mathbf{v}_{t-1}^\star)}$, 239
$g := \Delta\mu_t^\star/v$, and $\kappa_{\delta,t-1}$ is a sequence indexed by $t$ and de- 240
pending on $\delta$. Then, we switch from the observational to the 241
optimization phase in SADCBO when $\Delta\tilde{R}_t \leq s_t$, where 242

$$s_t := \frac{(\sigma_{t-1}(\mathbf{v}_t^\star) + \kappa_{\delta,t-1}/2)\sigma_{t-1}(\mathbf{v}_t)\sigma_{\text{noise}}\sqrt{-2\log\delta}}{\sigma_{t-1}^2(\mathbf{v}_t) + \sigma_{\text{noise}}^2}.$$

$$(12)$$

Further details about the derivation of $s_t$ and the expression 243
of $\kappa_{\delta,t-1}$ can be found in Appendix B. The entire algorithm 244
is summarized in Algorithm 1. 245

**Algorithm 1** SADCBO

---

1: **Input**: initial dataset $\mathcal{D}_0$, hyperparameters $\eta$ and $\gamma$, batch size $Q$, budget $\Lambda$, costs $\lambda_{\mathbf{x}}, \lambda_1, \ldots, \lambda_c$
2: Train initial GP using $\mathcal{D}_0$ and all variables $[\mathbf{x}, \mathbf{z}]$. phase = observational. $t = 1$.
3: **while** $\Lambda \geq \lambda_{\mathbf{x}}$ **do**
4:     Receive context $\mathbf{z}_{t+1} \sim p(\mathbf{z})$
5:     Assemble dataset $\mathcal{D}_t^{\text{BO}}$ (Equations (6) and (7))
6:     Compute $\text{FC}_{\mathcal{D}_t^{\text{BO}}}(j)$ for all $j$ (Equation (5) or (9) if phase = optimization)
7:     In descending order, add indices to $J_\eta$ until $\sum_{j \in J_\eta} \text{FC}_{\mathcal{D}_t^{\text{BO}}}(j) > \eta$
8:     Train lower-dimensional GP $\{(\mathbf{x}_i, \mathbf{z}_i^{(J_\eta)}, \mathbf{y}_i)\}_{i=1}^t$
9:     Get $\mathbf{x}_{t+1}$ (Equation (8)) (and $\mathbf{z}_{t+1}$ (Equation (10)) if phase = optimization)
10:    Acquire observation $y_{t+1}$ at $[\mathbf{x}_{t+1}, \mathbf{z}_{t+1}]$
11:    $\mathcal{D}_{t+1} \leftarrow \mathcal{D}_t \cup \{(\mathbf{x}_{t+1}, \mathbf{z}_{t+1}, y_{t+1})\}$
12:    Retrain full GP
13:    **if** phase = observational and $\Delta \tilde{R}_t \leq s_t$ [based on $p(f|\mathcal{D}_{t+1})$] (Equation (12)) **then**
14:       phase = optimization
15:    **end if**
16:    $\Lambda \leftarrow \Lambda - \lambda_{\mathbf{x}} + \sum_{j \in J_\eta} \lambda_j$, $t \leftarrow t+1$
17: **end while**

---

## 4 RELATED WORK

**Robust BO.** Bogunovic et al. [2018], Kirschner et al. [2020], Husain et al. [2023] and Saday et al. [2023] perform worst-case optimization under fluctuations of the contextual variables. In particular, Distributionally-Robust BO [DRBO, Kirschner et al., 2020] tries to maximize the expected black-box function value under the worst-case distribution of the contextual variables. This worst-case distribution belongs to an "uncertainty set", a ball centered around a reference distribution that is gradually learned [Tulabandhula and Rudin, 2014]. However, as in Krause and Ong [2011], these works assume that the relevant contextual variables are known *a priori*, and can only be observed, *after the designs have been selected*, and not controlled.

**High-dimensional BO.** Due to the curse of dimensionality, the performance of standard BO is severely degraded when applied in high-dimensional input spaces. To tackle this problem, most approaches either aim at carrying out BO in a lower-dimensional space instead of the original or work with a structured GP surrogate. A lower-dimensional subspace can be found in a data-agnostic manner, for instance by randomly dropping dimensions of the problem [Li et al., 2017] or considering tree-like random decompositions [Ziomek and Bou-Ammar, 2023]. Data-driven methods based on various measures of feature relevance have also been proposed [Spagnol et al., 2019, Shen and Kingsford, 2021]. In contrast, structured surrogate methods encode

structural information about the objective, for instance using an additive kernel, yielding an acquisition function that is additive under the provided decomposition [Rolland et al., 2018]. Finally, Eriksson and Jankowiak [2021] and Liu et al. [2023] proposed using a sparsity-enforcing GP surrogate, equipped with a heavy-tailed horseshoe prior on the squared inverse lengthscales.

**Cost-aware BO.** In most methods, the BO budget is given in iterations, implicitly assuming that each evaluation has the same cost. In practice, cost may vary significantly across different regions of the input space [Lee et al., 2020], or depend on the number of variables we optimize over. Cost-aware BO integrates the cost-constrained nature of the problem, usually within the acquisition function. Let us also mention more involved strategies like constrained Markov decision processes when the total budget is known beforehand [Lee et al., 2021]. The recent work by Tay et al. [2023] carries out Robust BO while at the same time involving a notion of controlled variables at a cost. However—unlike our framework—they require the nonselected variables to be sampled from a *known* distribution at each iteration.

## 5 EXPERIMENTAL RESULTS

We evaluate our approach on several real-world examples and synthetic functions, described in Section 5.1. We compare against multiple baselines (Table 1) and present results in Section 5.2. In Section 5.3 we discuss the influence of various experimental settings: number of noise variables present, contextual variable query cost, surrogate and method hyperparameters. We conclude by presenting several insights regarding the phase-switching criterion.

**Baselines.** We benchmark our approach, coined SADCBO, against baselines referenced in Table 1. In particular, MMDBO operates variable selection in BO through an MMD-based measure of sensitivity [Spagnol et al., 2019] and is detailed in Appendix C, whereas Dropout [Li et al., 2017] randomly selects half of the contextual variable for optimization. Next, CaBO [Lee et al., 2020] performs vanilla BO over $[\mathbf{x}, \mathbf{z}]$, using a cost-weighted acquisition function. The cost model employed here is a smoothed version of our non-continuous cost model, using a Gaussian curve. Finally, CBO refers to the Contextual BO framework proposed by Krause and Ong [2011]. As a way to assess the impact of contextual variables and selection mechanisms, we also report CUBO and VBO: Context-Unaware BO over the designs $\mathbf{x}$ only and Vanilla BO over both design and contextual variables $[\mathbf{x}, \mathbf{z}]$.

**Implementation details.** We fix the hyperparameters of SADCBO to $\eta = 0.8, Q = 10, \gamma_t = 0.8 \ \forall t$. For the GP surrogate, we use a squared-exponential kernel with independent lengthscales for each variable, learned through marginal likelihood maximization. We use the UCB acquisition strategy, as well as $Q$-UCB for computing $\mathcal{D}_t^Q$

Table 1: Methods used in experiments.

| | Name | Description |
|---|---|---|
| Without variable selection | CUBO | Context-Unaware BO over **x** only |
| | VBO | Vanilla BO over [**x, z**] |
| | CaBO | Cost-Aware BO over [**x, z**] [Lee et al., 2020] |
| | CBO | Contextual BO using all contexts **z** [Krause and Ong, 2011] |
| With variable selection | Dropout | Randomly drop half of context variables [Li et al., 2017] |
| | MMDBO | Maximum mean discrepancy-driven BO [Spagnol et al., 2019] |
| | SADCBO | **Sensitivity analysis-driven CBO (This work)** |

Table 2: Dimensionality of the experiments. For synthetic experiments, additional dimensions stand for (artificial) noise variables, put on top of the design and contextual variables.

| Experiment | All dimensions | Design variables | Contextual variables |
|---|---|---|---|
| Portfolio | 5 | 3 | 2 |
| Yacht | 6 | 4 | 2 |
| Robot | 14 | 6 | 8 |
| Molecule | 21 | 3 | 18 |
| EggHolder | 2 + 4 | 1 | 1 |
| Hartmann4D | 4 + 3 | 2 | 2 |
| Hartmann6D | 6 + 6 | 3 | 3 |
| Ackley | 5 + 8 | 2 | 3 |

(Equation (7)) [Wilson et al., 2017]. In all experiments, we assume that any variable, design or contextual ones, has cost $\lambda_j = 1 \; \forall j \in \{1, \dots, d + c\}$, except in a dedicated study in Section 5.3. Our algorithm is implemented using the `BoTorch` framework [Balandat et al., 2020]. Code can be accessed at https://github.com/julienmartinelli/SADCBO.

## 5.1 EXPERIMENTS

We benchmark on 4 real-world and 4 synthetic experiments (Table 2) described in brief here and detailed in Appendix E.

**Portfolio Optimization 5D.** This dataset was first introduced by Cakmak et al. [2020]. The goal is to optimize three design variables, which stand for the hyperparameters of a trading strategy, to maximize return under random environmental conditions. There are two contextual variables, namely: bid–ask spread and the borrowing cost.

**Yacht Hydrodynamics 6D.** This dataset comes from the UCI Machine Learning Repository [Gerritsma et al., 2013]. The optimization problem is to maximize the residuary resistance per unit weight of displacement of a yacht by controlling its 5-dimensional hull geometry coefficients. Design variables are the first four dimensions of the hull geometry coefficients. The contextual variables are the last hull geometry dimension and the Froude number.

**Molecule structure optimization 21D.** This computational chemistry example consists of optimizing the bond angles in an alanine molecule to determine the lowest energy conformer, i.e., the structure the molecule will likely take in nature. These problems are complicated by high dimensionality. We consider the Alanine, a molecule with 21 angular variables: 3 key variables based on prior domain knowledge set as design variables, and 18 other angles treated as contextual variables. Molecular energies are calculated with the AMBER forcefield [Case et al., 2023] at each round of BO.

**Robot pushing task 14D.** We follow Wang et al. [2017] and consider a control parameter tuning problem for robot pushing. This real-world function returns the distance between a designated goal location and two objects being pushed by two robot hands, whose trajectory is determined by 14 parameters specifying the location, rotation, velocity and moving direction, among others. There are 6 design variables and 8 contextual variables.

**Synthetic experiments.** We also consider four synthetic test functions, (see Table 2 and Appendix E.2 for details). A min-max transformation is performed on the input data, scaling it to the unit cube: $\mathcal{X} \times \mathcal{Z} = [0, 1]^{d+c}$. Similarly, the output is scaled between $[0, 1]$ and a noise term $\varepsilon \sim \mathcal{N}(0, \sigma_{\text{noise}}^2)$ is added with $\sigma_{\text{noise}}^2 = 0.001$. The contextual variable distribution is $p(\mathbf{z}) = \mathcal{U}([0, 1]^c)$.

## 5.2 RESULTS

**Real-world experiments.** In each plot from Figure 2, we report the best value found by each baseline as a function of the number of iterations. In real-world experiments (Figure 2a), SADCBO (in red with white markers) quickly converges to the optimum. SADCBO consistently outperforms the first baselines VBO and CUBO, even though in the Molecular Shape example, SADCBO and CUBO perform on par due to the good choices of the domain experts on the design variables. Except for the Robot Pushing task, the difference between SADCBO and CBO (in blue) is marginal in the real-world experiments. The latter enhances the surrogate model with information from sampled contexts, while our method may even optimize selected contextual variables if needed. Given that these baselines perform similarly, combined with the observation that optimizing only design variables (CUBO, in yellow) produces poor results for the Portfolio and Yacht problems, we can conclude that contextual variables play a significant part in maximizing these two objectives. The cost-aware BO baseline CaBO performs poorly in all tasks. Dropout and MMDBO consistently underperforms, except on the Yacht example for the latter. These baselines perform variable selection in a random manner for Dropout and using Hilbert-Schmidt Independence Criterion for MMDBO [Gretton et al., 2007], two strategies that do not seem to surpass the Feature Collapse method implemented in SADCBO. This observation highlights the need for an informed variable selection strategy. In-depth findings for the Molecule experiment are presented in Appendix E.1 and provide additional explanations as to why SADCBO clearly outperforms MMDBO and Dropout.

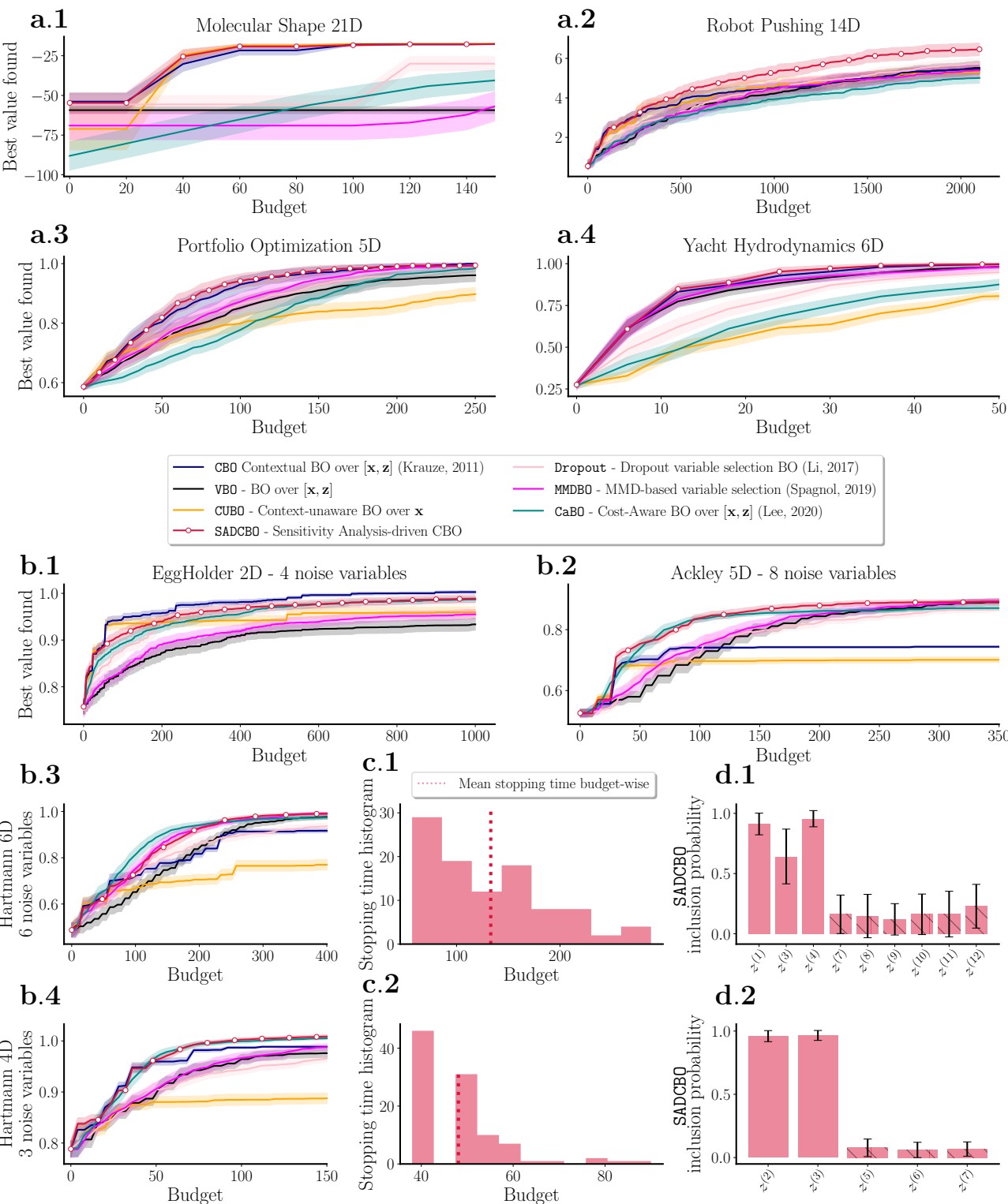

Figure 2: Benchmark of the different methods. **(a)** On real-world datasets, SADCBO (red curve with white markers) performs on par with other baselines and is the top performer for the Robot Pushing task. **(b)** On synthetic functions, SADCBO outperforms other baselines in three cases out of four. **(c)** Histograms of phase switching criteriong time for SADCBO computed for the Hartmann6D **(c.1)** and Hartmann4D problems **(c.2)**. **(d)** Inclusion probability of each contextual variable for SADCBO computed for the Hartmann6D **(d.1)** and Hartmann4D problems **(d.2)**. Each panel shows the mean $\pm 2$ standard error across $N = 100$ trials.

**Synthetic experiments.** Figure 2**b** displays the best value found by each baseline for synthetic functions. SADCBO ranks first on 3 out of 4 examples, closely followed by the cost-aware baseline CaBO, which performs much better on synthetic experiments than on the real-world ones. The contextual BO baseline CBO that obtained second to best results in real-world experiments, is now less performant, due to the fact that it does not optimize the context, similarly as CUBO. This seems to be particularly critical for Ackley5D, whereas for Hartmann6D/Hartmann4D, simply enhancing the surrogate with contextual variable observation already leads to a large performance gap between CUBO and CBO. Lastly, VBO does a poor job as it optimizes every variable, thus spending a large fraction of the budget every iteration.

For Hartmann6D and Hartmann4D, Figure 2**c** reports the time at which SADCBO's switching criterion (Equation (12)) kicks in, in proportion to the total budget, demonstrating that both phases are leveraged in our approach.

Finally, Figure 2**d** reports the sensitivity indices computed at each iteration for each contextual variable, averaged across whole trajectories of multiple trials. For Hartmann6D, the results match the Sobol sensitivity analysis results (Table S1), even though global sensitivity indices may differ from sensitivity indices with respect to the function optimum. Similar findings apply to Hartmann4D (Table S2). Results for other problems can be found in Figure S2.

> **Main takeaways.** Quantitatively, SADCBO achieves the best overall performances, ranking first in 7 out of 8 problems, although other methods obtained comparable performances on 5 out of 8 problems.
>
> The second-best and third-best methods, CBO and MMDBO, both severely underperform in two examples (Ackley and Hartmann6 for CBO, Molecular Shape and EggHolder for MMDBO). While the improvements provided by SADCBO may seem marginal, they are consistent across the benchmark.
>
> We hypothesize that this consistent behavior stems from our two-stage approach, which allows SADCBO to be versatile. SADCBO can handle both cases where the impact of the contextual variables on the function is limited (hence it is not worth spending budget to control them) and cases where spending budget leads to informative queries are simultaneously well-handled. For instance, SADCBO effectively reverted to a CBO algorithm in the Molecular Shape problem, due to an optimization phase mostly triggered at the end of the run. Meanwhile, for the Ackley function, the optimization phase was triggered in the first quarter of the budget on average, leading to SADCBO outperforming CBO.

## 5.3 SENSITIVITY ANALYSIS

We now report experiments assessing the robustness of SADCBO's performance to several modifications, either at the hyperparameter level or at the experiment setting level. The latter includes assessing performance when increasing the number of noise variables, varying the contextual variable query cost, or varying the surrogate model. Next, additional experiments illustrate the sound behavior of the proposed phase switching criterion implemented in SADCBO.

**Number of irrelevant contextual variables.** We compare the performance reached by SADCBO when adding an increasingly larger number of noise variables and find that even for a large number of irrelevant contextual variables, SADCBO reaches top performance on 3 out of 4 examples (Figure S3). The gap in performance between SADCBO and CaBO, Dropout and MMDBO seems to overall grow with the number of nuisance variables, in favor of SADCBO.

**Contextual variables optimization cost.** We investigate four different values for the query cost of contextual variables (Figure S4). For extremely cheap contextual variables $\lambda_j = 0.1$ for all $j$, that is, ten times cheaper than a design variable, VBO performs favorably, as optimizing over all inputs $[\mathbf{x}, \mathbf{z}]$ is cheap. SADCBO remains competitive in this configuration, even though MMDBO and CaBO perform on par. For a moderate cost $\lambda_j = 1$ (the cost model considered in Figure 2), SADCBO obtains the lowest average rank over all four test functions. For expensive contextual variables, $\lambda_j = 3$ or $\lambda_j = 10$, CaBO seems overall more suitable, although closely followed by SADCBO, and CBO.

**Sparsity-enforcing surrogates with SADCBO.** As SADCBO relies on a posterior sensitivity analysis to select the relevant contextual variables, and is hence agnostic to the choice of GP surrogate model, it can be combined with other methods that induce sparsity via the GP surrogate. One such method is by Eriksson and Jankowiak [2021], who introduced a sparsity-enforcing GP surrogate equipped with a horseshoe prior on the square inverse lengthscales, coined SAASBO. In Figure 3, we compare SADCBO with the combined method SAASBO+SADCBO, with both having the same hyperparameters. We observe that SAASBO+SADCBO improves over just SAASBO in all the synthetic examples, and is also better than SADCBO in two out of four examples. Note that the performance of SAASBO+SADCBO may further improve through hyperparameter tuning.

**SADCBO phase switching criterion.** We ensure that the criterion is well-behaved: the more information about the output is contained in the contextual variables, the later the phase switching occurs (Figure S5). Even though the stopping criterion was initially devised for vanilla BO, its application in a CBO setting is fruitful. Figure 4 further illustrates the soundness of the phase switching criterion.

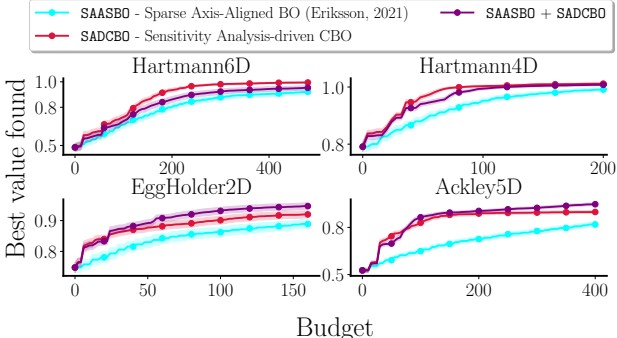

Figure 3: Combining `SADCBO` with sparsity-enforcing surrogate `SAASBO`. For any variable, the associated query cost is 1. $p(\mathbf{z}) = \mathcal{U}([0,1]^c)$. The combination is fruitful and improves the performances of `SAASBO`.

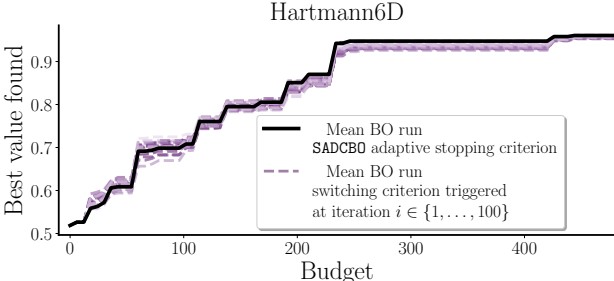

Figure 4: Assessing `SADCBO`'s phase switching criterion on the Hartmann6D function. The iteration selected by the adaptive stopping criterion implemented in `SADCBO` yields one of the best BO trials. Each curve is computed as an average of 10 different random seeds.

Using the Hartmann6D function under the same setting as described above, the mean switching iteration found by `SADCBO` over 100 different runs was collected. Then, new BO runs using `SADCBO` with a **fixed** phase switching time $i \in \{1, \dots, 100\}$ were performed. This was done 10 times for each switching time, using different random seeds for the initial dataset. The switching time found by `SADCBO` yields one of the best runs, validating the use of the criterion.

**`SADCBO` hyperparameters.** We vary the 3 hyperparameters of `SADCBO`: $\eta, \gamma, Q$. Unsurprisingly, the cumulative sensitivity threshold $\eta$ stands out as the most relevant parameter: as its value decreases, fewer variables are included, at which point not all relevant ones are selected, leading to reduced performance (Appendix D).

## 6 CONCLUSION

In this paper, we extended Contextual BO [Krause and Ong, 2011] to settings in which the contextual variables may be not only observed but also optimized at a cost. We intro-
duced `SADCBO`, an algorithm designed to select relevant context variables affecting the experimental outcomes by efficiently leveraging information present in both the observational and the interventional data. `SADCBO` results in more adequate surrogate models, and ensures the reproducibility of experiments by controlling for such relevant variables. In that respect, `SADCBO` should be used for practical applications where contextual variables can have an influence while being controllable. This includes, e.g., the development of new high-throughput materials or drugs, where machine learning strategies are being increasingly used [Zhang et al., 2020, Gómez-Bombarelli et al., 2018]. `SADCBO` can also be combined with any GP surrogate. Thus, if a practitioner believes that a specific contextual variable should be included, this can be easily achieved. Conversely, the variable selection procedure could be generalized to discard design variables as well. Lastly, recent work [Branchini et al., 2023] proposed to perform BO under the assumption that the input variables and the output are linked by a causal directed acyclic graph, learning the graph whilst maximizing the objective function. Despite its high computational complexity, applying this technique to our particular problem might be promising.

**Limitations and future work.** To achieve cost efficiency, `SADCBO` integrates the query cost at the variable selection level and employs an early stopping criterion. The latter only depends on an upper bound on the instantaneous regret difference and is therefore not cost-aware. Adding a notion of remaining budget to this criterion would certainly benefit our approach. On a similar note, while our algorithm incorporates cost, more effort could be put into specifying the costs. In our experiments, they were set to 1 for all variables to prevent bias in the results, and we carried out an ablation study with different costs in Section 5.3. Yet, it is worth mentioning that our method is compatible with the inference of black-box, input-dependent costs, similarly to `CaBO` [Lee et al., 2020]. One would simply need to modify Equation (9), replacing $\lambda_j$ by the learned cost. An interesting avenue for future work would be to elicit knowledge of experimental costs from domain experts in real-world situations.

## Acknowledgements

JM acknowledges the support of the Research Council of Finland under the HEALED project (grant 13342077). AB, AT, STJ, and PR were supported by the Research Council of Finland Flagship programme: Finnish Center for Artificial Intelligence FCAI. AT further acknowledges funding from the European Union's Horizon 2020 research and innovation programme under the Marie Skłodowska-Curie grant agreement No. 101059891. SJS and SK were supported by the UKRI Turing AI World-Leading Researcher Fellowship, [EP/W002973/1].

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

# Learning Relevant Contextual Variables Within Bayesian Optimization
## (Supplementary Material)

**Julien Martinelli**[*1]   **Ayush Bharti**[2]   **Armi Tiihonen**[3]   **S. T. John**[2]   **Louis Filstroff**[4]   **Sabina J. Sloman**[5]

**Patrick Rinke**[3]                              **Samuel Kaski**[2,5]

[1]Inserm Bordeaux Population Health, Vaccine Research Institute, Université de Bordeaux, Inria Bordeaux Sud-ouest, France
[2]Department of Computer Science, Aalto University, Helsinki, Finland
[3]Department of Applied Physics, Aalto University, Helsinki, Finland
[4]Univ. Lille, CNRS, Centrale Lille, UMR 9189 CRIStAL, F-59000 Lille, France
[5]Department of Computer Science, University of Manchester, Manchester, United Kingdom

# APPENDIX

**Outline.** The Appendix is organized as follows. In Appendix A, we provide a flowchart summarizing the proposed method SADCBO. In Appendix B, we provide further details about the phase switching criterion introduced in Section 3.2. In Appendix C, we provide more details about one of the baselines used in the main text, based on maximum mean discrepancy. Appendix D contains further experimental results regarding:

- Phase switching time and sensitivity-based inclusion probabilities of contextual variables found by SADCBO for additional test functions (Figure S2).
- Varying the number of irrelevant contextual variables (Figure S3).
- Varying contextual variables query cost (Figure S4).
- The distribution of phase switching times for SADCBO (Figure S5).
- Varying SADCBO hyperparameters (Appendix D.1 and Figure S6).

Finally, Appendix E contains a description of the real-world experiments performed throughout the paper, along with the analytical expressions of the synthetic examples used.

# A   FLOWCHART OF THE ALGORITHM

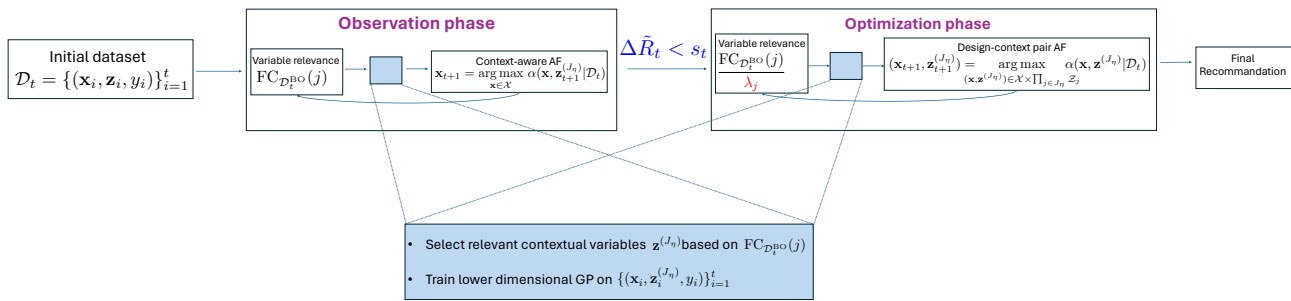

Figure S1: Flowchart of the proposed method SADCBO.

---

*Work done while at Aalto University.

## B   PHASE SWITCHING CRITERION

The phase switching criterion we employ is derived from the stopping criterion from Ishibashi et al. [2023]. The absolute difference of expected minimum simple regrets $\Delta R_t := |R_t - R_{t-1}|$ can be upper bounded with probability $1 - \delta$ by $\Delta \tilde{R}_t$, a quantity defined in Equation (11). Directly quoting the work of Ishibashi et al. [2023], the rationale behind this criterion reads as follows: "By evaluating the difference between the expected minimum simple regrets, we can stop BO without knowing $f^*$, because it indicates that the search efficiency is low and there is almost no improvement in the objective value. However, it is generally difficult to calculate $\Delta R_t$ analytically". Next, any stopping criterion involves the computation of some sort of threshold. Ishibashi et al. [2023] exploit the fact that their upper bound $\Delta \tilde{R}_t$ can itself be upper bounded by a quantity (introduced in [Ishibashi et al., 2023, Equation 10]), whose convergence speed to zero is limited by a specific term, $s_t$ (Equation 12). $s_t$ can be computed analytically and therefore yields an adaptive threshold.

Finally, Equation (11) involves a sequence $\kappa_{\delta, t-1}$:

$$\kappa_{\delta, t-1} = \max_{\mathbf{v} \in \mathcal{D}_{t-1}} \text{UCB}_\delta(\mathbf{v}) - \max_{\mathbf{v} \in \mathcal{V}} \text{LCB}_\delta(\mathbf{v}), \tag{S1}$$

where $\text{UCB}_\delta(\mathbf{v}) = \mu_t(\mathbf{v}|\mathcal{D}_t) + \beta_t^{1/2} \sigma_t(\mathbf{v}|\mathcal{D}_t)$ and $\text{LCB}_\delta(\mathbf{v}) = \mu_t(\mathbf{v}|\mathcal{D}_t) - \beta_t^{1/2} \sigma_t(\mathbf{v}|\mathcal{D}_t)$. $\beta_t^{1/2}$ is a trade-off parameter between exploration and exploitation that depends on $\delta$ [Srinivas et al., 2012]. $\kappa_{\delta, t-1}$ is a quantity that was first introduced by Makarova et al. [2022, Section 3.2] as an upper bound for the simple regret of the surrogate, which directly flows from the bounds provided by Srinivas et al. [2012] for well-calibrated surrogates.

Heuristically, one can think of our setting as applying the stopping criterion to $\mathbf{x} \mapsto f(\mathbf{x}, \mathbf{z})$, a stochastic black-box function with $\mathbf{z} \sim p(\mathbf{z})$. Upon satisfaction of this criterion, we switch to the optimization of $(\mathbf{x}, \mathbf{z}) \mapsto f(\mathbf{x}, \mathbf{z})$ where some contextual variables are optimized, and some others are still sampled from $p(z^{(j)})$.

## C   MAXIMUM MEAN DISCREPANCY-BASED VARIABLE SELECTION

Spagnol et al. [2019] introduced a BO algorithm with a variable selection procedure based on the Hilbert Schmidt Independence Criterion (HSIC). This measure can be used in our setting as well. We now briefly describe how it is defined.

As introduced in the main text, let $\mathcal{Z} \subset \mathbb{R}^c$ be the space of contextual variables, and $\mathcal{H}$ be a Hilbert space of $\mathbb{R}$-valued functions on $\mathcal{Z}$. Assume that $k : \mathcal{Z} \times \mathcal{Z} \to \mathbb{R}$ is the unique positive definite kernel associated with the Reproducing Kernel Hilbert Space $\mathcal{H}$. Let $\mu_{\mathbb{P}_Z}$ be the kernel mean embedding of the distribution $\mathbb{P}_Z$, $\mu_{\mathbb{P}_Z} := \mathbb{E}_Z[k(Z, \cdot)] = \int_{\mathcal{Z}} k(\mathbf{z}, \cdot) d\mathbb{P}_Z$. Kernel embeddings of probability measures provide a distance between distributions between their embeddings in the Hilbert Space $\mathcal{H}$, named Maximum Mean Discrepancy (MMD, [Gretton et al., 2012]):

$$\text{MMD}(\mathbb{P}_Z, \mathbb{P}_Y) = \|\mu_{\mathbb{P}_Z} - \mu_{\mathbb{P}_Y}\|_{\mathcal{H}}^2. \tag{S2}$$

For two random variables $Z \sim \mathbb{P}_Z$ on $\mathcal{H}$ and $Y \sim \mathbb{P}_Y$ on $\mathcal{G}$, the HSIC is the squared MMD between the product distribution $\mathbb{P}_{ZY}$ and the product of its marginals $\mathbb{P}_Z \mathbb{P}_Y$,

$$\text{HSIC}(Z, Y) = \text{MMD}^2(\mathbb{P}_{ZY}, \mathbb{P}_Z \mathbb{P}_Y) \tag{S3}$$

$$= \|\mu_{\mathbb{P}_{ZY}} - \mu_{\mathbb{P}_Z \mathbb{P}_Y}\|_{\mathcal{H} \otimes \mathcal{G}}^2 \tag{S4}$$

$$= \mathbb{E}_{Z,Y} \mathbb{E}_{Z',Y'}[k(Z, Z')l(Y, Y')] \tag{S5}$$

$$+ \mathbb{E}_Z \mathbb{E}_Y \mathbb{E}_{Z'} \mathbb{E}_{Y'}[k(Z, Z')l(Y, Y')]$$

$$- 2\mathbb{E}_{Z,Y} \mathbb{E}_{Z'} \mathbb{E}_{Y'}[k(Z, Z')l(Y, Y')].$$

To determine the relevance of a variable $Z^{(i)}$, Spagnol et al. [2019] introduce

$$S^{\text{HSIC}}(Z^{(i)}) = \text{HSIC}(Z^{(i)}, \mathbb{I}(Z \in \mathcal{L}_\gamma)), \tag{S6}$$

with $\mathcal{L}_\gamma$ a region of interest: the locations where the objective function value is above a threshold $\gamma$. This measure reflects how important $Z^{(i)}$ is to reach $\mathcal{L}_\gamma$.

We implemented this measure, substituting expectations for empirical means over the dataset $\mathcal{D}$. We use $\gamma = 0.8$, a threshold identical to the one used for SADCBO in Equation (6). The kernel $k$ is chosen to be a RBF kernel, and $l$ is a linear kernel $l(y, y') = yy'$, a common choice for binary data.

# D ADDITIONAL EXPERIMENTAL RESULTS

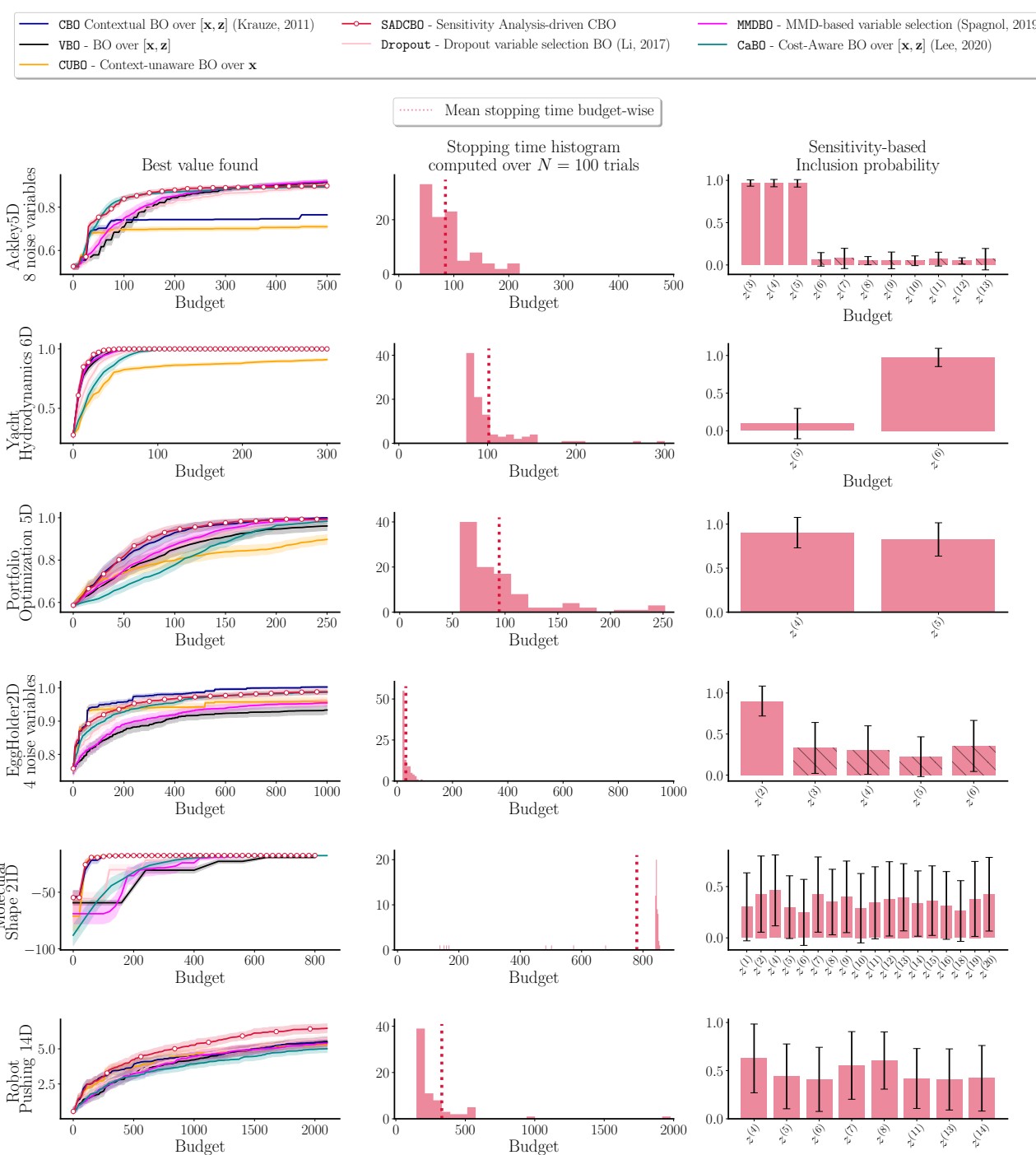

Figure S2: Each row deals with a specific problem. The left panel shows the BO trial results for each baseline. The middle and right panel show statistics related to SADCBO: 1) the phase switching time after which phase II begins and 2) the inclusion probabilities for each contextual variable. Statistics are computed across $N = 100$ BO trials with different seeds.

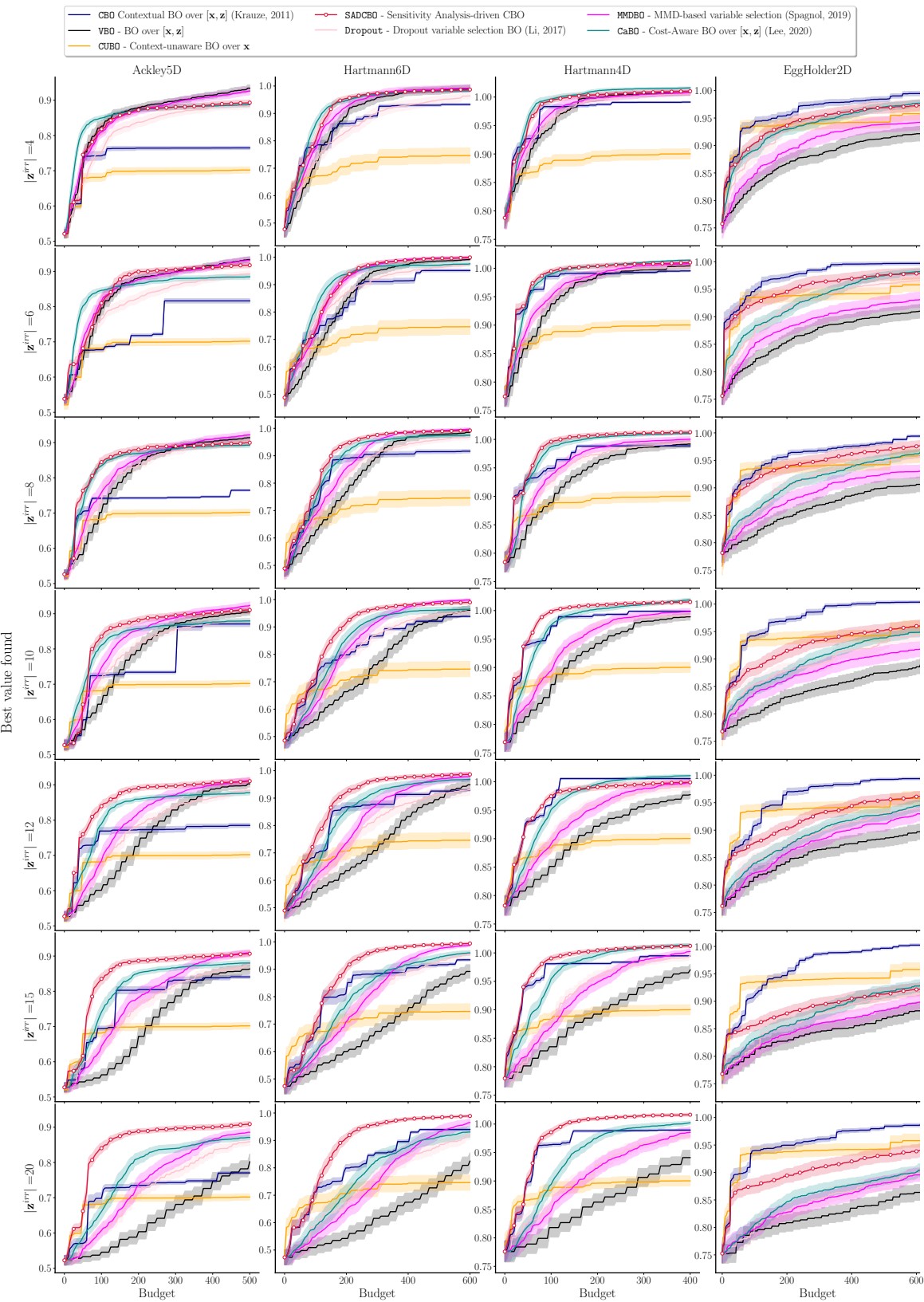

Figure S3: Varying the number of irrelevant contextual variables. For any variable, the associated query cost is 1. $p(\mathbf{z}) = \mathcal{U}([0,1]^c)$. On the three test functions Ackley5D, Hartmann6D and Hartmann4D, our approach outperforms other baselines even in high dimensions.

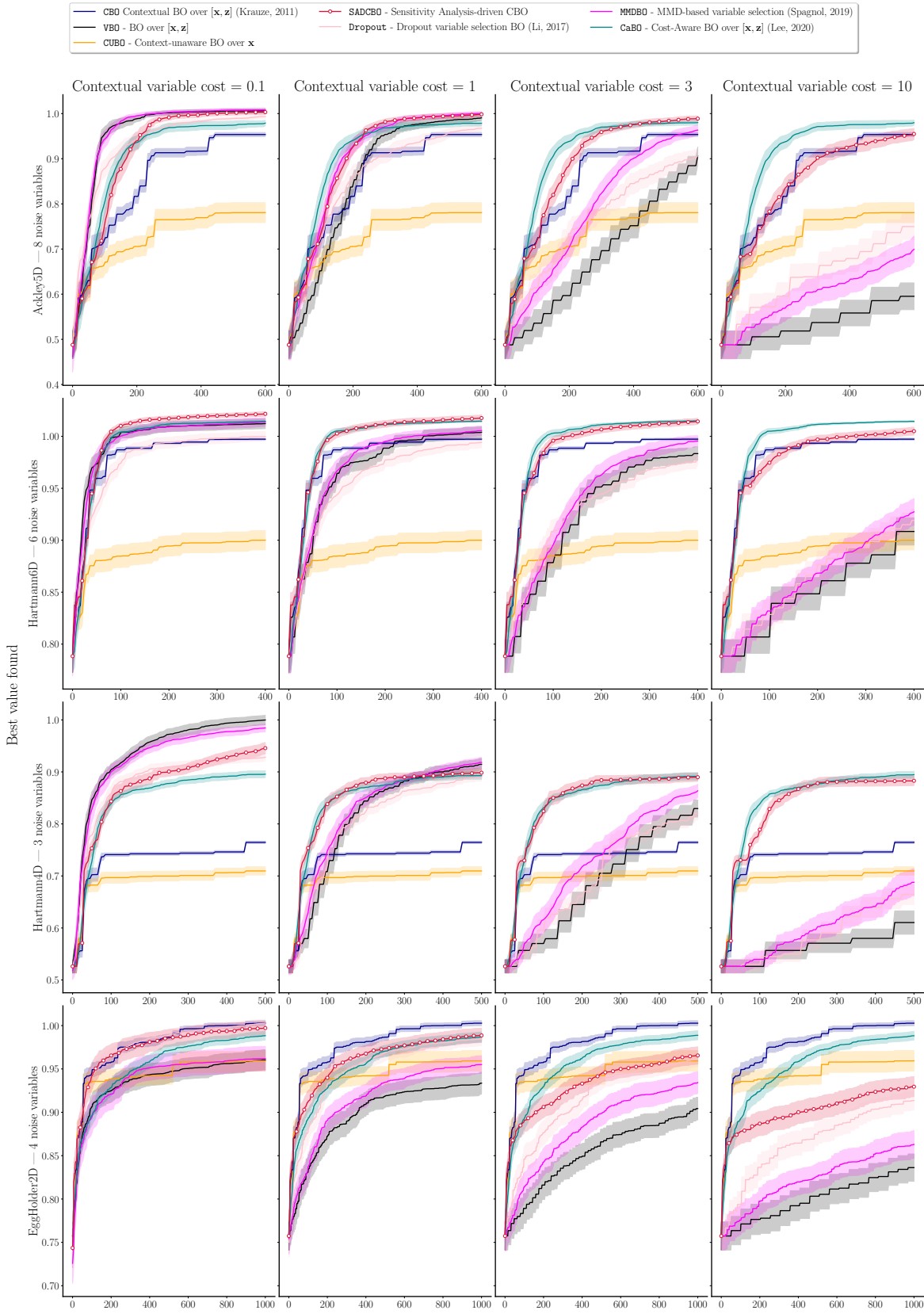

Figure S4: Ablation study on contextual variable query cost. Design variables have cost 1. $p(\mathbf{z}) = \mathcal{U}([0,1]^c)$.

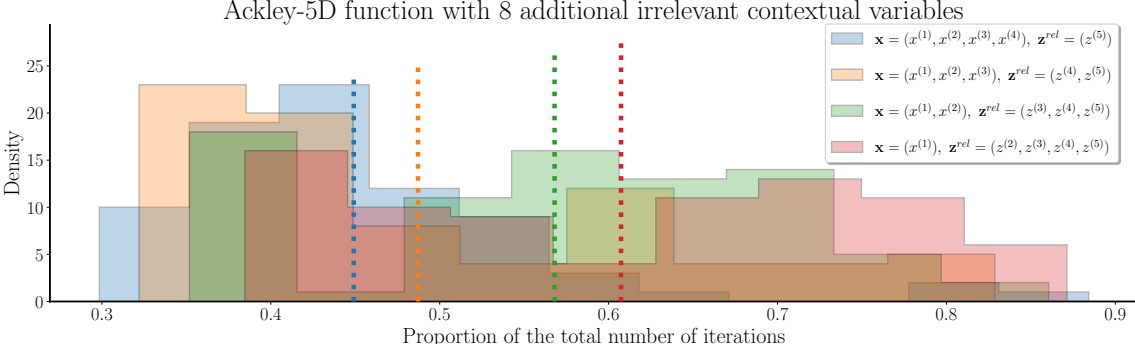

Figure S5: Distribution of phase switching criterion triggering times for `SADCBO` across $N = 100$ different BO trials. We consider the Ackley5D function with an increasingly larger ratio of relevant contextual variables over design variables, and 8 irrelevant contextual variables. $p(\mathbf{z}) = \mathcal{U}([0, 1]^c)$. For any variable, the associated query cost is 1. As the impact of contextual variables on the output function grows, the number of iterations spent in the observational phase grows as well.

## D.1 ADDITIONAL DETAILS ON HYPERPARAMETER VARIATIONS.

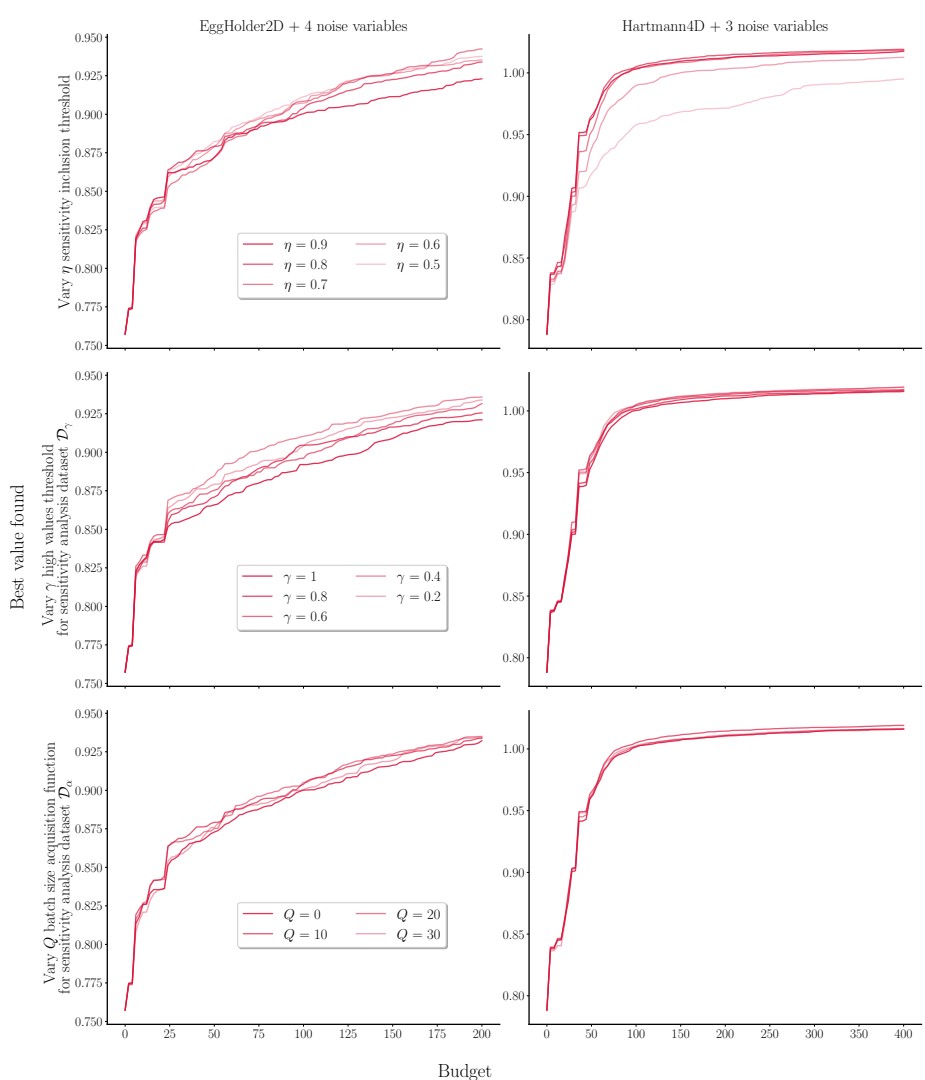

Figure S6: Varying hyperparameters for SADCBO. For any variable, the associated query cost is 1. $p(\mathbf{z}) = \mathcal{U}([0,1]^c)$. Top: varying $\eta$, the contextual variable inclusion threshold over the cumulative sum of sensitivity indices. Middle: varying $\gamma$, the threshold used in the creation of the truncated dataset $\mathcal{D}^\gamma$ from Equation (6). Bottom: varying $Q$, the size of the dataset $\mathcal{D}^Q$ from Equation (7). $\eta$ is the most sensitive hyperparameter here.

We vary the 3 hyperparameters of SADCBO: $\eta \in [0,1]$ the threshold based over the cumulative sum of sensitivity indices, which in turn regulates how many variables are selected every iteration; $\gamma \in [0,1]$, a threshold upon which a value is considered high enough to have its input added to dataset $\mathcal{D}^\gamma$ (Equation (6)), used for sensitivity analysis; and $Q$ the size of the dataset $\mathcal{D}^Q$ (Equation (7)).

Figure S6 reports the performances. Unsurprisingly, $\eta$ stands out as the most stringent parameter: as its value decreases, fewer variables are included, at which point not all relevant ones are selected, leading to reduced performances. Note that in a setting where there are no relevant contextual variables, lower values of $\eta$ will actually lead to better performances.Then, varying $\gamma \in [0,1]$ slightly affects the results: $\gamma$ increasing means that more samples are collected for sensitivity analysis, but these are less relevant for producing a reliable set of variables accounting for the fluctuations at the optimum. Finally, for the examples considered, $Q$ has only a limited effect, close to that of varying $\gamma$. This might stem from the fact that batched acquisition functions are notoriously difficult to optimize and may sometimes struggle to enforce diversity.

# E  EXPERIMENT DETAILS

## E.1  REAL-WORLD DATASETS

**Portfolio optimization dataset.**  This dataset was first introduced in [Cakmak et al., 2020]. The goal is to tune the hyper-parameters of a trading strategy so as to maximize return under risk-aversion to random environmental conditions. A software is used to simulate and optimize the evolution of a portfolio over a period of four years using open-source market data. Each evaluation of this simulator returns the average daily return over this period of time under the given combination of hyper-parameters and environmental conditions. Since the simulator is expensive to evaluate, we do not use it directly but perform pool-based Bayesian Optimization using a pool of 3000 points generated according to a Sobol sampling design.

The hyper-parameters to be optimized are the risk and trade aversion parameters and the holding cost multiplier. These variables constitute the design variables. The contextual variables are the bid-ask spread and the borrowing cost.

**Yacht hydrodynamics dataset.**  This dataset comes from the UCI Machine Learning Repository [Gerritsma et al., 2013]. The optimization problem is to maximize the residuary resistance per unit weight of displacement of a yacht by controlling its 5-dimensional hull geometry coefficients. Another optimization variable is the 1-dimensional Froude number. We chose as design variables the first four dimensions of the hull geometry coefficients. The contextual variables are the last hull geometry dimension and the Froude number. Like the Portfolio optimization dataset, we have access to a limited number of samples ($\approx 300$) and thus perform pool-based Bayesian optimization.

**Molecular structure optimization.**  This case is a computational chemistry challenge. Molecules can adopt different structures that preserve the topology (bonds and bonding types), but have different internal angles. Finding such conformers is a global optimization problem. Here, we are searching for the conformers of alanine — a molecule with structure $C_3H_7NO_2$ — whose energy is calculated at each round of BO with the AMBER force field [Salomon Ferrer et al., 2013, Case et al., 2023]. Alanine provides 33 structural variables to optimize: ten dihedral angles, eleven bond angles, and twelve bond lengths. Conformer search in the full 33-dimensional space is very challenging, but progress has been made with Bayesian optimization recently by reducing the problem to the four most important dihedral angles [Fang et al., 2021]. For the example in this work, three of these four dihedral angles were chosen as the design variables (indices 3, 17, and 21 in the dataset; which denotes dihedral angles d4, d11, and d13 in AMBER notation; d4 is the bond leading to the amino group, d13 the one leading to the hydroxyl group, and d11 is the bond between these two), the rest of the dihedral and bond angles (18 angles) are chosen as the contextual variables, and the bond lengths are kept fixed to facilitate faster simulations. The search space is selected by utilizing molecule domain knowledge in a conservative manner that allows 10-20 degree variations for the bond angles and is free for the dihedral angles. To outline the alanine optimization results, the structure optimization performed here as a test case is a high-dimensional problem, thus, the `VBO` method that tries to optimize all the variables **x** and **z** converges slowly. Due to the same reason, methods `MMDBO` and `Dropout` also underperform in terms of convergence for the alanine problem. Similarly to `SADCBO`, these two baselines operate variable selection, although using a different selection criterion. However, controlling the selected variable comes at a cost. On the opposite, it turns out that from Figure S2 (fifth row, middle panel), `SADCBO` virtually never switches to phase II for the Molecular Shape example. Therefore, `SADCBO` does perform contextual variable selection, but does not control them, it only chooses which of these variables will be included in the surrogate, hence behaving like `CBO`, but with a variable selection step. This explains why 1) `CBO` closely follows `SADCBO` for this example and 2) why other variable selection baselines like `MMDBO` and `Dropout` end up far from `SADCBO`. Interestingly, in this case, the simplified case of optimizing only the design variables **x** (`CUBO`) also performs well. This is because our domain experts made good initial choices on the relevant design variables **x** and the search spaces of context variables **z**. This type of pre-analysis is time-consuming and more challenging for larger molecules. Hence, a future line of work is to test context-aware BO more comprehensively in molecule structure optimization tasks.

**Robot Pushing Task**  This task was first introduced in Wang et al. [2017], and consists of a control parameter tuning problem for robot pushing. This real-world function returns the distance between a designated goal location and two objects being pushed by two robot hands, whose trajectory is determined by 14 parameters specifying the location, rotation, velocity and moving direction, among others. The function is implemented with a physics engine, the Box2D simulator. There are 6 design variables and 8 contextual variables.

### E.2 SYNTHETIC TEST FUNCTIONS

**Hartmann-6D function:**

$$f(\mathbf{v}) = -\sum_{i=1}^{4} \alpha_i \exp\left(-\sum_{j=1}^{6} A_{ij}(v^{(j)} - P_{ij})\right)$$

$$\alpha = (1.0, 1.2, 3.0, 3.2)^T$$

$$\mathbf{A} = \begin{pmatrix} 10 & 3 & 17 & 3.5 & 1.7 & 8 \\ 0.05 & 10 & 17 & 0.1 & 8 & 14 \\ 3 & 3.5 & 1.7 & 10 & 17 & 8 \\ 17 & 8 & 0.05 & 10 & 0.1 & 14 \end{pmatrix}$$

$$\mathbf{P} = 10^{-4} \begin{pmatrix} 1312 & 1696 & 5569 & 124 & 8283 & 5886 \\ 2329 & 4135 & 8307 & 3736 & 1004 & 9991 \\ 2348 & 1451 & 3522 & 2883 & 3047 & 6650 \\ 4047 & 8828 & 8732 & 5743 & 1091 & 381 \end{pmatrix}$$

defined over $\mathcal{V} = [0, 1]^6$. The second, fifth, and sixth variables were considered as design variables, while the first, third, and fourth variables were considered as contextual variables. 6 noise variables were added. Table S1 provides the results of a Sobol global sensitivity analysis performed using evaluations of the function collected over a grid of $N = 917504$ samples [Sobol, 2001]. Adding up the first order indices for design and contextual variables separately leads to $S_\mathbf{x} \approx 0.124$ and $S_\mathbf{z} \approx 0.196$. This means that with respect to first-order interactions, contextual variables have more impact than design variables, in this synthetic example. One should notice however that these indices are computed across the whole search space and not specifically at the optimum.

Table S1: Sobol global sensitivity analysis for the Hartmann-6D function using $N = 917504$ samples.

| Variable | First order sensitivity indices | Total order sensitivity indices |
|---|---|---|
| $z^{(1)}$ | 0.107 | 0.343 |
| $x^{(2)}$ | 0.006 | 0.399 |
| $z^{(3)}$ | 0.007 | 0.052 |
| $z^{(4)}$ | 0.082 | 0.379 |
| $x^{(5)}$ | 0.106 | 0.297 |
| $x^{(6)}$ | 0.012 | 0.482 |

**Hartmann-4D function:**

$$f(\mathbf{v}) = \frac{1}{0.839}\left(1.1 - \sum_{i=1}^{4} \alpha_i \exp\left(-\sum_{j=1}^{4} A_{ij}(v^{(j)} - P_{ij})\right)\right)$$

$$\alpha = (1.0, 1.2, 3.0, 3.2)^T$$

$$\mathbf{A} = \begin{pmatrix} 10 & 3 & 17 & 3.5 \\ 0.05 & 10 & 17 & 0.1 \\ 3 & 3.5 & 1.7 & 10 \\ 17 & 8 & 0.05 & 10 \end{pmatrix}$$

$$\mathbf{P} = 10^{-4} \begin{pmatrix} 1312 & 1696 & 5569 & 124 \\ 2329 & 4135 & 8307 & 3736 \\ 2348 & 1451 & 3522 & 2883 \\ 4047 & 8828 & 8732 & 5743 \end{pmatrix}$$

defined over $\mathcal{V} = [0, 1]^4$. The first and fourth variables were considered as design variables, while the second and third variables were considered as contextual variables. 3 noise variables were added. Table S2 provides the results of a Sobol

global sensitivity analysis performed using evaluations of the function collected over a grid of $N = 300000$ samples. Adding up the first order indices for design and contextual variables separately leads to $S_{\mathbf{x}} \approx 0.579$ and $S_{\mathbf{z}} \approx 0.091$. This means that with respect to first-order interactions, design variables have much more impact on the output than contextual variables. The gap slightly reduces when considering total order sensitivity indices. However, it is worth remembering that these indices are computed across the whole search space and not specifically at the optimum.

Table S2: Sobol global sensitivity analysis for the Hartmann-4D function using $N = 300000$ samples.

| Variable | First order sensitivity indices | Total order sensitivity indices |
|---|---|---|
| $x^{(1)}$ | 0.307 | 0.477 |
| $z^{(2)}$ | 0.037 | 0.279 |
| $z^{(3)}$ | 0.054 | 0.103 |
| $x^{(4)}$ | 0.272 | 0.526 |

**Ackley 5D function:**

$$f(\mathbf{v}) = -20 \exp\left(-0.2\sqrt{\frac{1}{5}\sum_{j=1}^{5}(v^{(j)})^2}\right) - \exp\left(\frac{1}{5}\sum_{j=1}^{5}\cos(2\pi v^{(j)})\right) + 20 + e^1$$

defined over $\mathcal{V} = [-5, 5]^5$. 8 noise variables were added.

**EggHolder 2D function:**

$$f(\mathbf{v}) = -(v^{(2)} + 47)\sin\left(\sqrt{\left|v^{(2)} + \frac{v^{(1)}}{2} + 47\right|}\right) - v^{(1)}\sin\left(\sqrt{|v^{(1)} - (v^{(2)} + 47)|}\right)$$

defined over $\mathcal{V} = [-512, 512]^2$. The first variable was considered as a design variable, and the second one as a contextual variable. 4 noise variables were added. A Sobol global sensitivity analysis performed using evaluations of the function collected over a grid of $N = 3000000$ samples shows that both variables have a similar contribution to the output (Table S3).

Table S3: Sobol global sensitivity analysis for the EggHolder-2D function using $N = 3000000$ samples.

| Variable | First order sensitivity indices | Total order sensitivity indices |
|---|---|---|
| $x^{(1)}$ | 0.001 | 0.998 |
| $z^{(2)}$ | 0.0004 | 0.999 |