# OpenReview forum: "Learning relevant contextual variables within Bayesian optimization"
_auai.org/UAI/2024/Conference — UAI 2024 poster_

### Official Review · Reviewer_4Cg4 · 2024-03-03

**Q2-1 Originality-Novelty:** 2
**Q2-2 Correctness-Technical Quality:** 3
**Q2-5 Clarity Of Writing:** 4

**Q1 Summary And Contributions:**

This paper proposes an algorithm for contextual bayesian optimization, when the context variables can be optimized at some additional cost. The algorithm has two stages, one in which the context variables are not controlled, and a second stage in which they are. The first observational stage is motivated by an assumption that only a subset of the context variables are actually relevant to the optimization problem.

Contributions include the new algorithm for this setting, and analysis of the algorithm in a lengthy set of experimental domains. Overall, the two-stage idea for this setting is an interesting one, but the need for this algorithm is not highlighted very clearly by experiments (in which at least one baseline compares favorably for each experimental setting), nor in theoretical analysis (with derivations relegated to the appendix). This motivates several of my questions for the authors, detailed in the boxes below.

**Q2-3 Extent To Which Claims Are Supported By Evidence:**

3: Good: the main claims are supported by convincing evidence (in the form of adequate experimental evaluation, proofs, (pseudo-)code, references, assumptions).

**Q2-4 Reproducibility:**

3: Good: key resources (e.g. proofs, code, data) are available and key details (e.g. proofs, experimental setup) are sufficiently well-described for competent researchers to confidently reproduce the main results.

**Q3 Main Strengths:**

Strengths of this paper include:
- The problem formulation that context variables could be optimized at some cost is an interesting one, with relevance to real-world problems, as discussed in this paper.
- The two-stage algorithm is clearly motivated by the assumptions of the problem setting.
- The number and variety of experimental domains is large. The sensitivity analyses add to the understanding of the relative performance of the proposed algorithm and baselines.
- The overall clarity of writing is high. Motivation is well scoped (with a short example in the intro), related work is summarized well, and care is taken to explain and contrast baseline methods (the table is particularly helpful for this).

**Q4 Main Weakness:**

A few points could be strengthened:
(1) I found the switching point between the observational stage and the control-of-context stage to need more motivation. Specifically, I understood the switching context in eq. (11) to apply when the marginal benefit from observation alone is small.
(1a) why is this a reasonable criteria for switching? In particular, considering just marginal benefit of another observation round does not account for the possibility that observation rounds could still be valuable, but round of controlling the context variables could be even more valuable?
(1b) why are the stages in this order (first observe, then control?)
(1c) It would be great if the authors could give some interpretation of the key terms in eq. 11, especially since the derivation is entirely in the appendix.
(1d) does the derivation of the algorithm point to the settings in which this approach will be most fruitful,

(2) While there are many experimental domains and sensitivity analyses, none of them seem to really emphasize strengths of the proposed algorithm.*
(2a) I found the assessment "SADCBO ranks first on 3 out of 4 examples, closely followed by the cost-aware baseline CaBO" a bit misleading, as the two are pretty much indistinguishable in the experiment figure.
(2b) the proposed algorithm is at best a minor improvement over baselines for the real datasets
(2c) what can we take away from these results? Maybe that for these domains, controlling the context variables is not necessary (in which case, maybe focusing on one more real-world motivated domain where it does would be stronger)? Or maybe that this two-stage algorithm or the switching conditions is not a significant improvement over the baselines?
* please let me know during the rebuttal period if I'm interpreting the results in fig. 2 incorrectly.

**Q5 Detailed Comments To The Authors:**

My main questions are in the "weaknesses" box above.

**Q9 Complying With Reviewing Instructions:**

Yes

---

> ### Author Rebuttal · Authors · 2024-04-04
>
> *Why is SADCBO's two-phase algorithm in an observe-first-then-optimize
> order rather than the opposite?*
>
> Informally, in the early part of the trial, every query provides a lot
> of information because of the limited number of points in the dataset.
> Hence, even with "cheap" queries, where not all variables are controlled
> for, a lot of information is acquired. Once many queries have been
> gathered, information becomes scarce, hence controlling more dimensions
> of the problem for a significantly higher price becomes necessary. This
> reasoning is quite similar to the mechanisms found in the Multi-Fidelity
> BO literature, where one is allowed to leverage cheap "low-fidelity"
> information sources to aid the maximization of a target function. In
> this framework, the algorithms usually begin by exploiting the
> low-fidelity sources at low cost, and then only later perform expensive
> queries to the target function.
>
> *What are the main takeaways from the synthetic and real-world
> experiments?*
>
> -   From a quantitative point of view, SADCBO achieves the best overall
>     performances, ranking first in 7 out of 8 problems, even though
>     other methods obtained similar performances for 5 out of 7 problems.
>
> -   The next two best methods are CBO and MMDBO, both severely
>     underperforming in two examples (Ackley and Hartmann6 for CBO,
>     Molecular Shape and EggHolder for MMDBO). While the improvements
>     provided by SADCBO may seem marginal, they are consistent across the
>     whole benchmark.
>
> -   We hypothesize that this consistent behavior stems from our
>     two-stage approach, which allows SADCBO to be versatile. Indeed,
>     cases where contextual variables's impact on the function is limited
>     (and hence not worth spending budget controlling them) and cases
>     where spending budget leads to informative queries are
>     simultaneously well-handled. As an example, the former was
>     illustrated by the Molecular Shape experiment, where SADCBO
>     effectively reverted to a CBO algorithm as the optimization phase
>     was mostly triggered only at the end of the BO run. The latter can
>     be seen for instance with the Ackley Function, where the
>     optimization phase is quickly triggered (in the first quarter of the
>     budget on average), which allows SADCBO to quickly outperform CBO.
>
> ----
>
> Please also see the general comments.

---

### Official Review · Reviewer_B8rS · 2024-03-21

**Q2-1 Originality-Novelty:** 3
**Q2-2 Correctness-Technical Quality:** 3
**Q2-5 Clarity Of Writing:** 3

**Q1 Summary And Contributions:**

They authors tackles a problem of CBO where the cost of contextual variables varies and current methods simply include context variables as part of the design.

Instead, they "adaptively select a subset of contextual variables to include in the optimization, based on the trade-off between their relevance and the addi- tional cost incurred by optimizing them compared to leaving them to be determined by the environ- ment."

They effectively “extend the CBO framework to settings in which the relevance of contextual variables is (i) not known beforehand, and can be optimized at some cost”.

**Q2-3 Extent To Which Claims Are Supported By Evidence:**

3: Good: the main claims are supported by convincing evidence (in the form of adequate experimental evaluation, proofs, (pseudo-)code, references, assumptions).

**Q2-4 Reproducibility:**

3: Good: key resources (e.g. proofs, code, data) are available and key details (e.g. proofs, experimental setup) are sufficiently well-described for competent researchers to confidently reproduce the main results.

**Q3 Main Strengths:**

Problem has clear merit for the CBO literature.

The chosen approach is clear and effective to demonstrate their idea.

Their execution is complete, carrying all the necessary arguments and evidence to make their case complete.

**Q4 Main Weakness:**

None that I can see.

**Q5 Detailed Comments To The Authors:**

Questions:
a) There is significant conceptual overlap to the causal BO literature, e.g. in the conclusion it says: “select relevant context variables affecting the experimental outcomes by efficiently leveraging information present in both the ob- servational and the interventional data.”

Effectively, selecting relevant context variables is an identical task in causal BO, and so a comparison between the two needs to be done, though, due to space constraints I expect this to be covered in another paper in the future.

b) Fig 2a1 and 2a2: Is it possible to get a bit more insight on why SADCBO is outperforming here, especially in 2a2?

For 2a1, it seems more or less identical to CBO or CUBO, but it clearly stands out in 2a2

Typos:
a) “Design 352 variables  CONTROL the first four dimensions of the hull geometry coef-353 ficients.”

**Q9 Complying With Reviewing Instructions:**

Yes

---

> ### Author Rebuttal · Authors · 2024-04-04
>
> *Relation to the causal BO literature*
>
> We agree that the setting introduced in this paper can be viewed under
> the lens of causality. The work of Branchini et al. \[2023\], where one
> aims to carry BO while simultaneously leveraging the causal but unknown
> dependencies between variables, may apply to our setting. **This will be
> mentioned in the discussion**.
>
> *Insight on better performances of SADCBO in Figure 2.a.1 and 2.a.2*
>
> Regarding the Molecular Shape experiment (Figure 2.a.1), detailed
> explanations were provided in the Appendix due to space constraints
> (Page 17, L753). These explanations read as follows: To outline the
> alanine optimization results, the structure optimization performed here
> as a test case is a high-dimensional problem, thus, the VBO method that
> tries to optimize all the variables $\mathbf{x}$ and $\mathbf{z}$
> converges slowly. Due to the same reason, methods MMDBO and Dropout also
> underperform in terms of convergence for the alanine problem. Similarly
> to SADCBO, these two baselines operate variable selection, although
> using different selection criterion. However, controlling the selected
> variable comes at a cost. On the opposite, it turns out that from Figure
> S2 (fifth row, middle panel), SADCBO virtually never switches to phase
> II for the Molecular Shape example. Therefore, SADCBO does perform
> contextual variable selection, but does not control them, it only
> chooses which of these variables will be included in the surrogate,
> hence behaving like CBO, but with a variable selection step. This
> explains why 1) CBO closely follows SADCBO for this example and 2) why
> other variable selection baselines like MMDBO and Dropout end up far
> from SADCBO. Interestingly, in this case, the simplified case of
> optimizing only the design variables $\mathbf{x}$ (CUBO) also performs
> well. This is because our domain experts made good initial choices on
> the relevant design variables x and the search spaces of context
> variables $\mathbf{z}$. This type of pre-analysis is time-consuming and
> more challenging for larger molecules. Hence, a future line of work is
> to test context-aware BO more comprehensively in molecule structure
> optimization tasks.
>
> About the Robot Pushing experiment (Figure 2.a.2), one can also get
> further insights on SADCBO's performances using Figure S2 in the
> Appendix. There, the mean switching time between phases is reported
> across different repetitions (bottom row, middle panel). It can be seen
> that the switching time happens quite early in the BO trial, on average
> at one-fifth of the total budget. Next, contextual variables inclusion
> probabilities (bottom row, right panel) show that among the 8 contextual
> variables, there is no clear distinction, with all contextual variables
> being included roughly half of the time. This means that SADCBO selected
> on average 4 contextual variables to optimize. As a comparison, the
> baseline MMDBO selected an average number of 5 contextual variables to
> optimize, thus spending more budget at every iteration. Overall, the use
> of one-fifth of the budget for the observation phase jointly with a
> reduced number of optimized variables yields a cost-efficient use of the
> budget by SADCBO, compared to MMDBO. This might explain SADCBO's better
> performances.
>
> ----
>
> Please also see the general comments.

---

### Official Review · Reviewer_KHJx · 2024-03-22

**Q2-1 Originality-Novelty:** 3
**Q2-2 Correctness-Technical Quality:** 3
**Q2-5 Clarity Of Writing:** 2

**Q10 Ethical Concerns:**

No.

**Q1 Summary And Contributions:**

In contextual Bayesian optimization, high-dimensional contextual vectors are used in the process of Bayesian optimization without any additional processing.  However, there can be effective and ineffective variables in the contextual vectors.  In this paper, the authors tackle the problem of identifying such effectiveness of contextual variables using the feature collapsing method.  Finally, the authors show the experimental results to demonstrate the validity of the proposed method.

**Q2-3 Extent To Which Claims Are Supported By Evidence:**

3: Good: the main claims are supported by convincing evidence (in the form of adequate experimental evaluation, proofs, (pseudo-)code, references, assumptions).

**Q2-4 Reproducibility:**

2: Fair: key resources (e.g. proofs, code, data) are unavailable but key details (e.g. proof sketches, experimental setup) are sufficiently well-described for an expert to confidently reproduce the main results.

**Q3 Main Strengths:**

- It solves an interesting problem in contextual Bayesian optimization.
- Some components (such as the feature collapsing method and early termination strategy of Bayesian optimization) included in the proposed method are not that novel, but the authors utilize them appropriately.
- Numerical analysis supports the authors’ claims.

**Q4 Main Weakness:**

- The authors can improve the presentation and writing of the paper.

**Q5 Detailed Comments To The Authors:**

- I think that the proposed method solves the problem of interest appropriately, but the authors need to clarify and highlight technical contributions.
- Presentation and writing can be improved more.  For example, in Line 141, `adaption` should be `adaptation`, and in Line 203, `We found FC to perform better, see Section 5.2` seems not connected to the previous sentences.  I don’t describe all typos and poor writing, but I think the authors can improve the paper by proof-reading it carefully.
- You can improve some figures.  For example, in Figure 1, I am hard to distinguish between two lines in the bottom plots.  Moreover, in Figure 3, tick labels are overlapped to other figures.  You need to fix them.
- It is a minor thing, but the title should be a title case; `Learning Relevant Contextual Variables within Bayesian Optimization`.

**Q9 Complying With Reviewing Instructions:**

Yes

---

> ### Author Rebuttal · Authors · 2024-04-04
>
> *Clarification on technical contributions*
>
> Our technical contributions can be listed as follows:
>
> -   We present the problem of optimizing a black-box function involving
>     potentially irrelevant but costly contextual variables, thus
>     suggesting the existence of a tradeoff between relevance and
>     optimization cost, requiring a specific treatment.
>
> -   Towards that end, we adapt the Feature Collapsing method, a variable
>     selection method for Gaussian Processes based on sensitivity
>     analysis. We tailor it to the Bayesian Optimization case, where one
>     is mostly interested in the impact of a variable on the *maximum*
>     value of a function rather than over its whole domain. This leads us
>     to consider a modified version of the dataset over which feature
>     relevance is computed.
>
> -   We introduce a novel algorithm that operates in a two-phase manner.
>     Beginning with an *observation phase*, contextual variables are not
>     optimized for, only observed as realizations of the environment.
>     Yet, a subset of them is integrated into the surrogate, based on the
>     abovementioned variable selection procedure. This leads to cheap
>     queries that are informative enough, given that we are in the early
>     stage of the BO trial. Once significant improvements in the best
>     value found cannot be achieved anymore, we enter the *optimization
>     phase*, where contextual variable can now be optimized for. This
>     leads to more informative queries, but at a cost. As such, the
>     variable selection procedure now takes into account contextual
>     variables cost, thus providing a cost-efficient set of contexts to
>     optimize for. One switches from the first to the second stage once a
>     stopping criterion is triggered. The latter is an adaptation of the
>     recent work of Ishibashi *et al.* \[2023\].
>
> For a comprehensive summary of the algorithm, we refer to the flowchart
> accessible at this anonymous link:
> https://anonymous.4open.science/r/UAI2024RebuttalBO-6833/ .
>
> ----
>
> Please also see the general comments.

---

### Official Review · Reviewer_CqKm · 2024-03-22

**Q2-1 Originality-Novelty:** 3
**Q2-2 Correctness-Technical Quality:** 2
**Q2-5 Clarity Of Writing:** 2

**Q10 Ethical Concerns:**

No.

**Q1 Summary And Contributions:**

This article is to solve the problem of contextual variables in Bayesian Optimization (BO), especially when the correlation of these variables is unknown in advance and optimizing them requires additional costs.
 The SADCBO algorithm combines sensitivity analysis driven variable selection and early stopping criteria to improve optimization performance while maintaining cost efficiency. Through extensive ablation studies in synthesis and real-world experiments, it has been demonstrated that the SADCBO method has improved consistency in multiple examples compared to existing methods.

**Q2-3 Extent To Which Claims Are Supported By Evidence:**

2: Fair: the main claims are somewhat supported by evidence (but the experimental evaluation may be weak, or does not match entirely with the claims, important baselines may be missing, proofs contain important ideas but lack rigor, algorithmic details are only discussed superficially, references are imprecise, assumptions are not sufficiently motivated or explicated, etc.).

**Q2-4 Reproducibility:**

2: Fair: key resources (e.g. proofs, code, data) are unavailable but key details (e.g. proof sketches, experimental setup) are sufficiently well-described for an expert to confidently reproduce the main results.

**Q3 Main Strengths:**

The SADCBO algorithm selects variables based on sensitivity analysis, which has a solid theoretical foundation in statistics and machine learning. The design of the algorithm takes into account the cost efficiency during the optimization process and demonstrates good performance in experiments.
Through experiments on synthetic and real-world datasets, the author has demonstrated the advantages of SADCBO over other baseline methods in multiple test cases. These experimental results support the author's claim that SADCBO can effectively handle contextual variable related issues. The structure of the paper is reasonable and the logic is clear. Firstly, the background and motivation of the problem are introduced, followed by a detailed description of the algorithm design and experimental results. Finally, the advantages of the algorithm and potential areas for improvement are discussed.

**Q4 Main Weakness:**

Although the experimental results show good performance of SADCBO, these experiments may require further validation, especially in different types of problems and broader application scenarios.
Although the authors provided experimental evidence to support their claims, these experiments may need to be conducted on more datasets and different contextual variable settings to further demonstrate the algorithm's generalization ability
Before the paper is accepted and the code is publicly available, other researchers may not be able to fully reproduce the experimental results, which may affect the initial evaluation of algorithm performance.

**Q5 Detailed Comments To The Authors:**

Consider adding an intuitive chart or diagram to help readers better understand the working principle of the SADCBO algorithm and its differences from other algorithms.
When introducing the advantages of algorithms, more examples or case studies can be used to specifically demonstrate the performance of SADCBO in practical applications.
 Ensure that all relevant work cited is listed in the references so that readers can further explore and validate the proposed method.
More literature on the application of sensitivity analysis in Bayesian optimization may need to be cited to strengthen the theoretical foundation of this method.
Carefully proofread the entire text to ensure that there are no missed minor errors, spelling errors, or grammar issues. These minor issues may distract readers and affect the overall quality of the paper.
For some complex sentences, it is possible to consider simplifying the grammar structure to make the sentences clearer and easier to understand.

How does the SADCBO algorithm perform when dealing with large-scale or high-dimensional problems? Is there any optimization or improvement for this type of problem?
How does SADCBO adapt and handle different cost structures for cost models in practical applications? Is there a specific strategy to handle cost changes?

**Q9 Complying With Reviewing Instructions:**

Yes

---

> ### Author Rebuttal · Authors · 2024-04-04
>
> *Additional experiments on high-dimensional problems*
>
> First, it is worth noticing that we already considered somewhat
> high-dimensional problems in the paper, when an ablation study on the
> number of irrelevant contextual variables was conducted. There, we
> experimented with as many as 20 irrelevant contextual variables and
> found SADCBO to demonstrate competitive performance (Figure S3),
> achieving the best ranking in 3 out of 4 synthetic problems. However,
> this high-dimensional case involved mostly irrelevant contextual
> variables. We now present the results for two high-dimensional synthetic
> problems where a significant number of variables have nonzero relevance,
> i.e., they have a contribution to the function output. These can be
> accessed through the anonymous link
> https://anonymous.4open.science/r/UAI2024RebuttalBO-6833/ (Figure 1). We
> considered the Ackley and Rosenbrock functions, each with 30 relevant
> dimensions and 10 irrelevant contextual variables. The relevant
> variables are split into 20 design variables $\mathbf{x}_{1:20}$
>
>  and 10
> contextual variables
>
> $\mathbf{z}_{21:30}$
>
>  for each function. Irrelevant
> contextual variables are $\mathbf{z}_{31:40}$. Our results show that
> SADCBO is still competitive in this scenario, achieving the best ranking
> on the Ackley function. For the Rosenbrock function, most baselines
> perform on par and SADCBO is competitive, converging to the optimum
> nearly as fast as the top performer Cost-Aware BO.
>
> ----
>
> Please also see the general comments.

---

### Official Review · Reviewer_QRmS · 2024-03-23

**Q2-1 Originality-Novelty:** 3
**Q2-2 Correctness-Technical Quality:** 3
**Q2-5 Clarity Of Writing:** 3

**Q1 Summary And Contributions:**

The paper addresses the challenges of Bayesian optimization (BO) in high-dimensional input spaces and the varying costs associated with evaluating different variables. It introduces SADCBO, an extension of Contextual BO that optimizes both observed and controllable contextual variables at a cost. Key contributions include efficient selection of relevant contextual variables, improved surrogate modeling, and ensuring experiment reproducibility. The method's flexibility allows for practical applications in domains like material or drug development, while acknowledging limitations such as the need for cost-awareness and better specification of costs for real-world scenarios.

**Q2-3 Extent To Which Claims Are Supported By Evidence:**

3: Good: the main claims are supported by convincing evidence (in the form of adequate experimental evaluation, proofs, (pseudo-)code, references, assumptions).

**Q2-4 Reproducibility:**

3: Good: key resources (e.g. proofs, code, data) are available and key details (e.g. proofs, experimental setup) are sufficiently well-described for competent researchers to confidently reproduce the main results.

**Q3 Main Strengths:**

- The incorporation of sensitivity analysis-driven variable selection and cost-aware optimization represents an interesting advance over the current state-of-the-art in BO.
-  The two-phase strategy with observation and optimization phases, leveraging cost information and early stopping criteria, is a novel contribution to CBO.
- SADCBO integrates cost-aware optimization, considering the varying costs associated with different variables. By optimizing both design and contextual variables at a cost, SADCBO offers a more realistic and practical approach to optimization in real-world scenarios.
-  The paper presents comprehensive experiments with comparisons to baseline methods, demonstrating SADCBO's ability to perform well, especially regarding relevant contextual variables and cost considerations.
- The paper is well-written, and the experimental results are clearly presented (via several figures). Key resources (algorithm, data) are available and key details (e.g. experimental setup) are sufficiently well-described, authors mention that the code will be available upon acceptance.

**Q4 Main Weakness:**

- The paper uses cost-aware BO methods but lacks a rigorous explanation of how cost information is incorporated into the Gaussian Process (GP). A more detailed explanation would strengthen the technical aspects.

- The paper highlights the significance of costs in real-world applications; however, it does not explore methodologies for estimating or eliciting these costs from domain experts (This point was adressed in the conclusion).

**Q5 Detailed Comments To The Authors:**

- Can you elaborate on the specific mathematical formulation used to incorporate cost information into the Gaussian Process (GP) surrogate model? Does it involve cost-weighted kernels or cost-modulated acquisition functions?
- How does SADCBO handle situations where the cost of manipulating a contextual variable is unknown or uncertain? Are there any robustness measures implemented to deal with such scenarios?
The paper proposes an upper bound on the instantaneous regret difference for early stopping. However, even with the explanation provided in Appendix A, the intuition behind the stopping criterion, specifically why ∆R˜_t ≤ s_t is chosen, remains unclear. Additionally, a more in-depth explanation of equations 10 and 11 would be beneficial for a better understanding.
- Could you provide more details on the algorithmic complexity of SADCBO and are there any potential optimizations or modifications that could enhance its efficiency for large-scale experiments?

- The description of the databases after Table 2 seems a bit long.  Since the table already has the important information,  authors can use that space to share some interesting findings from Appendix A instead

**Q9 Complying With Reviewing Instructions:**

Yes

---

> ### Author Rebuttal · Authors · 2024-04-04
>
> *Could you provide more details on the algorithmic complexity of SADCBO
> and are there any potential optimizations or modifications that could
> enhance its efficiency for large-scale experiments?*
>
> The computationally heavy part of SADCBO is captured by Equation 4,
> which assesses a contextual variable's relevance by its induced change
> in posterior predictive distribution when this variable's value is
> turned to 0 at a given point. This requires computing a KL divergence
> for each contextual variable, and each point of dataset
> $\mathcal{D}^{\text{BO}}_t$. Although this computation involves two
> Gaussian distributions and therefore has a closed-form, it requires us
> to perform a prediction at numerous points, a task which has complexity
> $\mathcal{O}(N^3)$ for $N$ the number of observations, when using for
> vanilla GPs. Efficiency improvements can be achieved by considering
> Sparse GPs with inducing points, for instance, or approximations like
> the Nyström method. **These potential improvements will be included as
> part of the Discussion.**
>
> ----
>
> Please also see the general comments.

---

### Meta-Review · Area_Chair_Kxe5 · 2024-04-15

The paper proposes a method for Bayesian Optimisation to select a subset of contextual variables to include in the optimisation, based on the trade-off between their relevance and the additional cost incurred by optimising them compared to leaving them fixed. Reviewers agree that this is a solid paper and well written. The authors have convincingly replied to the Reviewers questions.